# Disruption of thalamic functional connectivity is a neural correlate of dexmedetomidine-induced unconsciousness

Oluwaseun Akeju[1]*[†], Marco L Loggia[2,3]*[†], Ciprian Catana[2], Kara J Pavone[1], Rafael Vazquez[1], James Rhee[1], Violeta Contreras Ramirez[1], Daniel B Chonde[2], David Izquierdo-Garcia[2], Grae Arabasz[2], Shirley Hsu[2], Kathleen Habeeb[4], Jacob M Hooker[2], Vitaly Napadow[2,3], Emery N Brown[1,5,6], Patrick L Purdon[1,6]*

[1]Department of Anesthesia, Critical Care and Pain Medicine, Massachusetts General Hospital, Harvard Medical School, Boston, United States; [2]MGH/MIT/HMS Athinoula A Martinos Center for Biomedical Imaging, Charlestown, United States; [3]Department of Anesthesiology, Perioperative and Pain Medicine, Brigham and Women's Hospital, Harvard Medical School, Boston, United States; [4]Clinical Research Center, Massachusetts General Hospital, Boston, United States; [5]Department of Brain and Cognitive Science and Institute for Medical Engineering and Sciences, Massachusetts Institute of Technology, Cambridge, United States; [6]Harvard-Massachusetts Institute of Technology Division of Health Sciences and Technology, Massachusetts Institute of Technology, Cambridge, United States

**\*For correspondence:**
oluwaseun.akeju@mgh.harvard.edu (OA); marco@nmr.mgh.harvard.edu (MLL); patrickp@nmr.mgh.harvard.edu (PLP)

[†]These authors contributed equally to this work

**Competing interests:** The authors declare that no competing interests exist.

**Reviewing editor**: Jody C Culham, University of Western Ontario, Canada

**Abstract** Understanding the neural basis of consciousness is fundamental to neuroscience research. Disruptions in cortico-cortical connectivity have been suggested as a primary mechanism of unconsciousness. By using a novel combination of positron emission tomography and functional magnetic resonance imaging, we studied anesthesia-induced unconsciousness and recovery using the α2-agonist dexmedetomidine. During unconsciousness, cerebral metabolic rate of glucose and cerebral blood flow were preferentially decreased in the thalamus, the Default Mode Network (DMN), and the bilateral Frontoparietal Networks (FPNs). Cortico-cortical functional connectivity within the DMN and FPNs was preserved. However, DMN thalamo-cortical functional connectivity was disrupted. Recovery from this state was associated with sustained reduction in cerebral blood flow and restored DMN thalamo-cortical functional connectivity. We report that loss of thalamo-cortical functional connectivity is sufficient to produce unconsciousness.

## Introduction

Understanding the neural mechanisms of consciousness is a fundamental challenge of current neuroscience research. We note that precise definitions of the words 'consciousness' and 'unconsciousness' remain ambiguous. However, loss of voluntary response in a task is an effective and clinically relevant definition of unconsciousness. Over the past decade, important insights have been gained by using functional magnetic resonance imaging (fMRI) to characterize the brain during sleep, and disorders of consciousness (DOC) such as coma, vegetative states, and minimally conscious states (*Greicius et al., 2008*; *Horovitz et al., 2008*; *Boly et al., 2009*; *Cauda et al., 2009*; *Horovitz et al., 2009*; *Larson-Prior et al., 2009*; *Vanhaudenhuyse et al., 2010*; *Samann et al., 2011*; *Fernandez-Espejo et al., 2012*; *Picchioni et al., 2014*). In addition to being critical for patients to safely and humanely undergo

**eLife digest** Although we are all familiar with the experience of being conscious, explaining precisely what consciousness is and how it arises from activity in the brain remains extremely challenging. Indeed, explaining consciousness is so challenging that it is sometimes referred to as 'the hard question' of neuroscience.

One way to obtain insights into the neural basis of consciousness is to compare patterns of activity in the brains of conscious subjects with patterns of brain activity in the same subjects under anesthesia. The results of some experiments of this kind suggest that loss of consciousness occurs when the communication between specific regions within the outer layer of the brain, the cortex, is disrupted. However, other studies seem to contradict these findings by showing that this communication can sometimes remain intact in unconscious subjects.

Akeju, Loggia et al. have now resolved this issue by using brain imaging to examine the changes that occur as healthy volunteers enter and emerge from a light form of anesthesia roughly equivalent to non-REM sleep. An imaging technique called PET revealed that the loss of consciousness in the subjects was accompanied by reduced activity in a structure deep within the brain called the thalamus. Reduced activity was also seen in areas of cortex at the front and back of the brain.

A technique called fMRI showed in turn that communication between the cortex and the thalamus was disrupted as subjects drifted into unconsciousness, whereas communication between cortical regions was spared. As subjects awakened from the anesthesia, communication between the thalamus and the cortex was restored.

These results suggest that changes within distinct brain regions give rise to different depths of unconsciousness. Loss of communication between the thalamus and the cortex generates the unconsciousness of sleep or light anesthesia, while the additional loss of communication between cortical regions generates the unconsciousness of general anesthesia or coma. In addition to explaining the mixed results seen in previous experiments, this distinction could lead to advances in the diagnosis of patients with disorders of consciousness, and even to the development of therapies that target the thalamus and its connections with cortex.

traumatic surgical or diagnostic procedures, anesthesia has long been recognized as a tool for studying the neural mechanisms of loss and recovery of consciousness (*Beecher, 1947*).

Disruption of cortico-cortical connectivity has been proposed as a central mechanism to explain unconsciousness associated with general anesthesia, sleep, and DOC (*Massimini et al., 2005*; *Mashour, 2006*; *Alkire et al., 2008*; *Boly et al., 2009*; *Cauda et al., 2009*; *Boveroux et al., 2010*; *Vanhaudenhuyse et al., 2010*; *Langsjo et al., 2012*; *Jordan et al., 2013*). However, recent evidence suggests that cortico-cortical functional connectivity may be maintained during unconsciousness (*Greicius et al., 2008*; *Horovitz et al., 2008*; *Larson-Prior et al., 2009*). To develop more precise neuroanatomic and neurophysiological characterizations of this and other putative mechanisms, studies of anesthetic-induced unconsciousness could benefit from the use of multimodal imaging approaches, combined with neurophysiological understanding of the brain state induced by the anesthetic drug being studied.

A challenge in using anesthetics to study unconsciousness is stating precise, possible mechanisms of anesthetic-induced brain states, and using an anesthetic which acts at single rather than multiple brain targets to test those mechanism (*Rudolph and Antkowiak, 2004*; *Franks, 2008*; *Brown et al., 2011*). Among the several anesthetics in current use today, dexmedetomidine is an appealing choice for testing a specific mechanism of how an anesthetic alters the level of consciousness. Dexmedetomidine selectively targets pre-synaptic $\alpha_2$-adrenergic receptors on neurons projecting from the locus ceruleus to the pre-optic area (*Correa-Sales et al., 1992*; *Chiu et al., 1995*; *Mizobe et al., 1996*). This leads to activation of inhibitory outputs to the major arousal centers in the midbrain, pons, and hypothalamus producing a neurophysiological and behavioral state that closely resembles NREM II sleep (*Figure 1*) (*Nelson et al., 2003*; *Huupponen et al., 2008*; *Akeju et al., 2014*). Dexmedetomidine also acts at the locus ceruleus projections to the intralaminar nucleus of the thalamus, the basal forebrain and the cortex (*Espana and Berridge, 2006*).

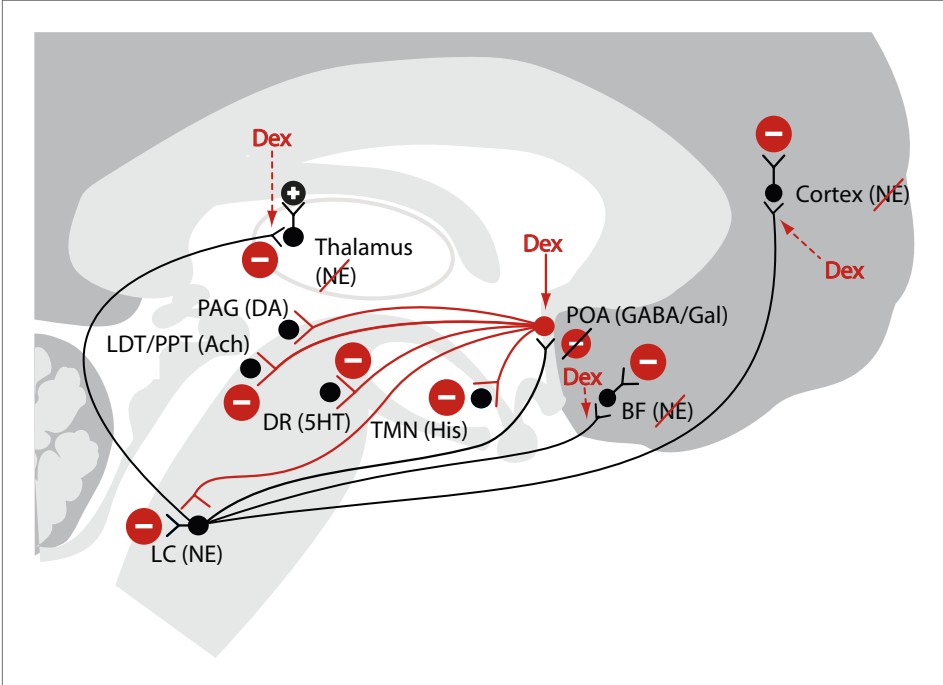

**Figure 1**. Schematic of dexmedetomidine signaling illustrating similarities to the mechanism proposed for the generation of non-rapid eye movement II sleep. Dexmedetomidine binds to α2 receptors on neurons emanating from the locus ceruleus to inhibit NE release in the POA. The disinhibited POA reduces arousal by means of GABA- and galanin-mediated inhibition of the midbrain, hypothalamic, and pontine arousal nuclei. Dexmedetomidine also acts at the locus ceruleus projections to the intralaminar nucleus of the hypothalamus, the basal forebrain and the cortex and on post-synaptic α2 receptors. 5HT, serotonin; Ach, acetylcholine; BF, basal forebrain; DA, dopamine; Dex; dexmedetomidine; DR, dorsal raphe; GABA, gamma aminobutyric acid receptor subtype A; Gal, galanin; His, histamine; LC, locus ceruleus; LDT, laterodorsal tegmental area; NE, norepinephrine; PAG, periaqueductal gray; POA, preoptic area; PPT, pedunculopontine tegmental area; TMN, tuberomamillary nucleus.

Therefore, the aim of this study was to use a novel integrated positron emission tomography and magnetic resonance imaging (PET/MR) approach to characterize brain resting-state network activity and metabolism during dexmedetomidine-induced unconsciousness. This strategy offers several appealing features. First, PET scanning using fludeoxyglucose ($^{18}$F-FDG) provides a sensitive estimate of the brain's metabolic state in terms of the cerebral metabolic rate of glucose ($CMR_{glc}$). Second, comparison of $CMR_{glc}$ results with regional cerebral blood flow (rCBF) estimates obtained from pulsed Arterial Spin Labeling (pASL) fMRI signals allows a direct assessment of whether, as previously reported, flow-metabolism coupling is maintained during dexmedetomidine-induced unconsciousness (*Drummond et al., 2008*). Third, we use the $CMR_{glc}$ results to inform the brain functional connectivity analysis derived from the Blood Oxygen Level Dependent (BOLD) fMRI signals.

## Results

### Decreased CMRglc in the Default Mode Network (DMN), Fronto Parietal Networks (FPNs), and thalamus during dexmedetomidine-induced unconsciousness

In 10 healthy volunteers, we used EEG recordings, and an auditory task to confirm that dexmedetomidine induced a loss of voluntary responsiveness, and exhibited a neurophysiological profile that was similar to non rapid eye movement (NREM) II sleep (*Akeju et al., 2014*). We then studied $^{18}$F-FDG uptake during baseline and dexmedetomidine-induced unconsciousness in these 10 healthy volunteers in a separate experiment where we defined loss of consciousness as the onset of sustained eye closure and lack of response to a verbal request to open the eyes (*Figure 2A*). During both the baseline and dexmedetomidine-induced unconsciousness $^{18}$F-FDG study visits (two per subject), fMRI data

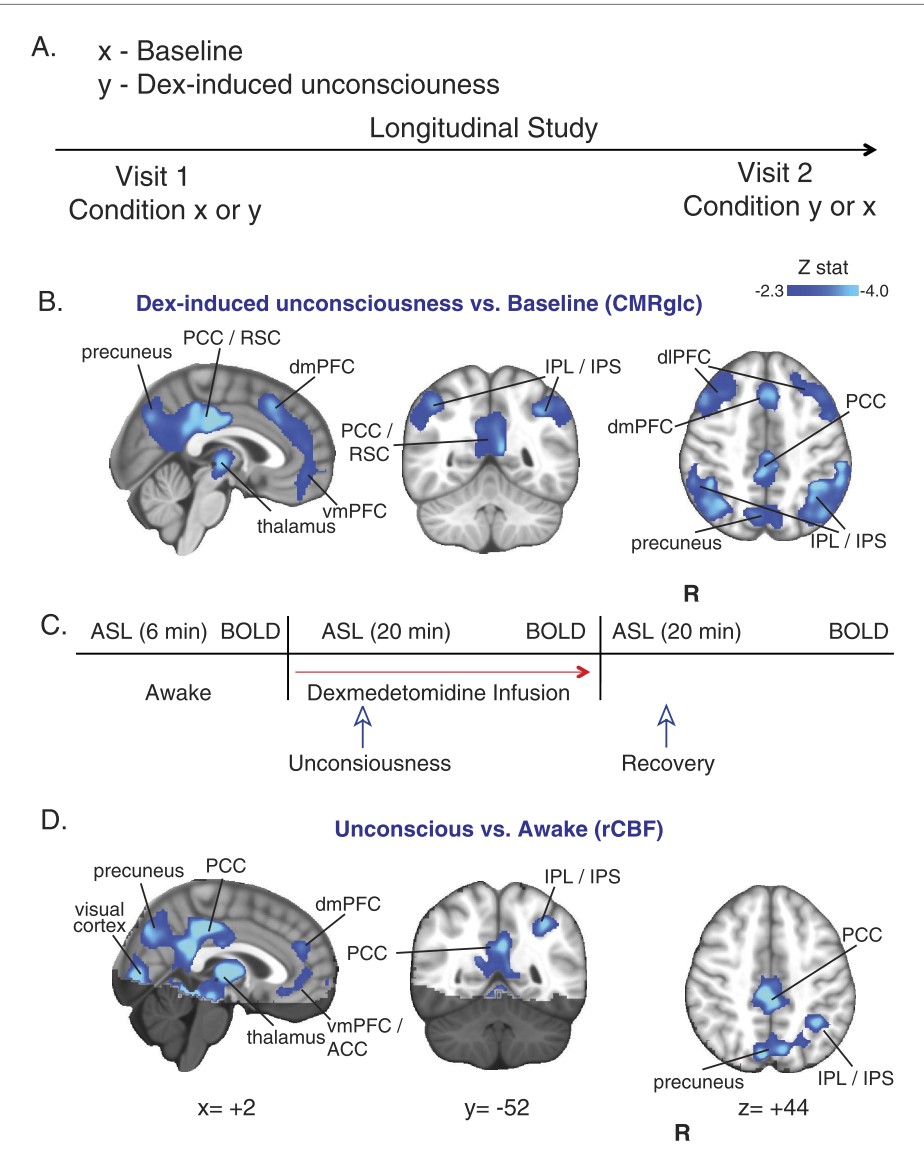

**Figure 2**. CMR$_{glc}$ and rCBF are decreased in both the default mode and the frontal–parietal network brain regions during unconsciousness. (**A**) Study outline of healthy volunteers recruited to undergo both baseline and dexmedetomidine-induced unconsciousness PET brain imaging in a random order. (**B**) Group wise (n = 10) dexmedetomidine-induced unconsciousness vs baseline changes in CMR$_{glc}$ (PET) displayed on the MNI152 standard volume. Significant CMR$_{glc}$ decreases were localized to brain regions that make up the Default Mode and the Frontal-Parietal Networks. (**C**) Schematic of fMRI obtained concurrently in combined PET/MR visits (n = 10) and MR/only visits (n = 7). (**D**) Group wise (n = 17) dexmedetomidine-induced unconsciousness vs awake changes in rCBF (fMRI) displayed on the MNI152 standard volume. Significant rCBF decreases were also localized to brain regions that make up the Default Mode and the Frontal-Parietal Networks. The brain regions that were not included in the rCBF estimation are shaded in the darker hue. ACC, anterior cingulate cortex; CMR$_{glc}$, cerebral metabolic rate of glucose; Dex, dexmedetomidine; dmPFC, dorsomedial prefrontal cortex; fMRI, functional magnetic resonance; IPL, infraparietal lobule; IPL, infraparietal sulcus; PCC, posterior cingulate cortex; R, right; rCBF, regional cerebral blood flow; RSC, restrosplenial cortex; vmPFC, ventromedial prefrontal cortex.

The following figure supplement is available for figure 2:

**Figure supplement 1**. PET SUV changes during unconsciousness.

were also recorded. Consistent with a global decrease in glucose metabolism, we found that during the unconscious state, there was a broad reduction in $^{18}$F-FDG standardized uptake values in the brain (*Figure 2—figure supplement 1A*). To quantify this result, we calculated CMR$_{glc}$ during the two $^{18}$F-FDG visits (*Figure 2—figure supplement 1B*). We found reduced CMR$_{glc}$ in thalamic, frontal, and parietal brain regions during the unconscious state (*Figure 2B*, *Table 1*). The difference in CMR$_{glc}$ exhibited a spatial distribution consistent with the previously described DMN (*Raichle et al., 2001*; *Damoiseaux et al., 2006*) and both left and right FPNs (*Table 1*) (*Damoiseaux et al., 2006*; *Vincent et al., 2008*). We found these network-specific CMR$_{glc}$ changes interesting because the DMN has been linked to stimulus-independent thought and self-consciousness (*Raichle et al., 2001*; *Mason et al., 2007*; *Vincent et al., 2008*), while the FPNs have been linked to conscious perception of the external environment and executive control initiation (*Boly et al., 2008*; *Vincent et al., 2008*).

## Decreased rCBF in DMN, FPN, and thalamus during dexmedetomidine-induced unconsciousness

To further understand the effects of dexmedetomidine-induced unconsciousness on rCBF and brain functional connectivity in these networks, we increased our statistical power for fMRI data analysis by

**Table 1.** CMRglc, awake vs dex-induced unconsciousness

| Label | Z-stat | MNI coordinate (mm) | | | cluster size (# voxels) | cluster p-value |
| | | x | y | z | | |
|---|---|---|---|---|---|---|
| awake > dex-induced unconsciousness | | | | | | |
| R middle/inferior frontal gyrus | 4.35 | 52 | 42 | 4 | 17919 | 2.95E-12 |
| R medial orbital gyrus | 4.22 | 24 | 38 | −16 | | |
| L inferior frontal gyrus | 4.02 | −52 | 36 | 10 | | |
| L superior frontal gyrus, medial part | 4 | −6 | 20 | 40 | | |
| L middle frontal gyrus | 3.94 | −42 | 18 | 38 | | |
| L middle frontal gyrus | 3.86 | −34 | 8 | 66 | | |
| R middle frontopolar gyrus | 3.79 | −30 | 58 | 2 | | |
| L lateral orbital gyrus | 3.58 | −46 | 28 | −12 | | |
| L medial orbital gyrus | 3.56 | −26 | 32 | −22 | | |
| R middle frontal gyrus | 3.55 | 48 | 20 | 44 | | |
| R superior frontal gyrus | 3.48 | 26 | 2 | 56 | | |
| R inferior frontal gyrus, pars opercularis | 3.37 | 54 | 16 | 4 | | |
| L inferior frontal gyrus, pars opercularis | 3.36 | −46 | 20 | 0 | | |
| L superior frontal gyrus, medial part (2) | 3.33 | −4 | 42 | 22 | | |
| inferior rostral gyrus | 3.15 | 0 | 48 | −10 | | |
| R anterior insula | 2.82 | 40 | 22 | 0 | | |
| L Posterior Cingulate Gyrus | 5.56 | −4 | −24 | 38 | 5443 | 2.85E-05 |
| L precuneus | 4.27 | −12 | −64 | 22 | | |
| R precuneus | 3.86 | 12 | −60 | 28 | | |
| L supramarginal gyrus | 4.58 | −40 | −46 | 42 | 2861 | 0.0029 |
| L angular gyrus | 3.76 | −48 | −68 | 34 | | |
| R supramarginal gyrus | 4.59 | 44 | −46 | 48 | 2214 | 0.0112 |
| R angular gyrus | 4 | 34 | −62 | 44 | | |
| R thalamus | 4.52 | 12 | −20 | 4 | 1610 | 0.0444 |
| L thalamus | 3.89 | −14 | −26 | 4 | | |

CMRglc, cerebral metabolic rate of glucose; Dex, dexmedetomidine; L, left; MNI, Montreal Neurological Institute; R, right.

studying an additional seven volunteers using fMRI only (*Figure 2C*). Analysis of the rCBF difference maps between the unconscious and baseline states showed decreases in rCBF during unconsciousness that overlapped with the same DMN and FPN regions that demonstrated reductions in CMR$_{glc}$ (*Figure 2D*, *Figure 2—figure supplement 1*, *Table 2*). This is consistent with maintenance of blood flow and metabolism coupling during dexmedetomidine-induced unconsciousness.

## Decreased thalamo-cortical and cortico-cerebellar functional connectivity during dexmedetomidine-induced unconsciousness

Given that we observed reduced flow and metabolism in regions known to correspond to the DMN and FPNs (*Raichle et al., 2001*; *Damoiseaux et al., 2006*; *Vincent et al., 2008*), we next asked whether intrinsic functional connectivity within these networks was altered with dexmedetomidine-induced unconsciousness. First, we confirmed that during both wakefulness and unconsciousness, the DMN and bilateral FPNs were consistently identified in our volunteers (*Figure 3*, *Table 3*). Next, we performed a comparison of changes in these networks between the unconscious and awake states. When we compared the DMN between these two states, we found no difference in functional connectivity between the cortical regions of the DMN and FPNs, suggesting that cortico–cortico functional connectivity may be maintained during unconsciousness (*Figure 3A–B*, *Table 3*). However, we observed a reduction in functional connectivity between the thalamus and the DMN in the unconscious state (*Figure 3A–B*, *Table 3*). This loss of thalamic functional connectivity was observed in a region consistent with intralaminar, midline, mediodorsal, and ventral anterior nuclei. However, our ability to precisely resolve the thalamic nuclei implicated in loss of thalamo-cortical functional

**Table 2.** rCBF, awake vs unconscious

| Label | Z-stat | MNI coordinate (mm) x | y | z | cluster size (# voxels) | cluster p-value |
|---|---|---|---|---|---|---|
| awake > unconscious | | | | | | |
| L thalamus | 5.36 | −10 | −14 | 12 | 10876 | 4.20E-11 |
| L midbrain | 5.2 | 8 | −32 | −10 | | |
| L posterior cingulate cortex | 5.14 | −2 | −46 | 28 | | |
| R thalamus | 5.03 | 8 | −12 | 10 | | |
| R midbrain | 4.44 | −6 | −36 | −12 | | |
| L precuneus | 4.44 | −2 | −70 | 32 | | |
| R precuneus | 4.3 | 8 | −76 | 42 | | |
| R intracalcarine cortex | 4.2 | 4 | −84 | 4 | | |
| L intracalcarine cortex | 4.18 | −2 | −82 | 2 | | |
| R hippocampus | 4.17 | 30 | −34 | −4 | | |
| L supramarginal gyrus/intraparietal sulcus | 4.03 | −34 | −50 | 44 | | |
| L hippocampus | 3.09 | −26 | −22 | −12 | | |
| R frontopolar cortex | 4.71 | −24 | 64 | 2 | 1773 | 0.00463 |
| L frontopolar cortex | 4.32 | 20 | 62 | −6 | | |
| L middle frontal gyrus | 3.77 | −22 | 38 | 28 | | |
| R inferior frontal gyrus, orbital part | 2.88 | 50 | 40 | −14 | | |
| L dorsal anterior cingulate cortex | 3.93 | 10 | 38 | 22 | 1262 | 0.0227 |
| R dorsal anterior cingulate cortex | 3.75 | −10 | 38 | 20 | | |
| L subgenual anterior cingulate cortex | 3.3 | −2 | 24 | −10 | | |
| R pregenual anterior cingulate cortex | 3.21 | 6 | 42 | −6 | | |
| unconscious > awake | | | | | | |
| n.s. | | | | | | |

L, left; MNI, Montreal Neurological Institute; n.s, not significant; R, right.

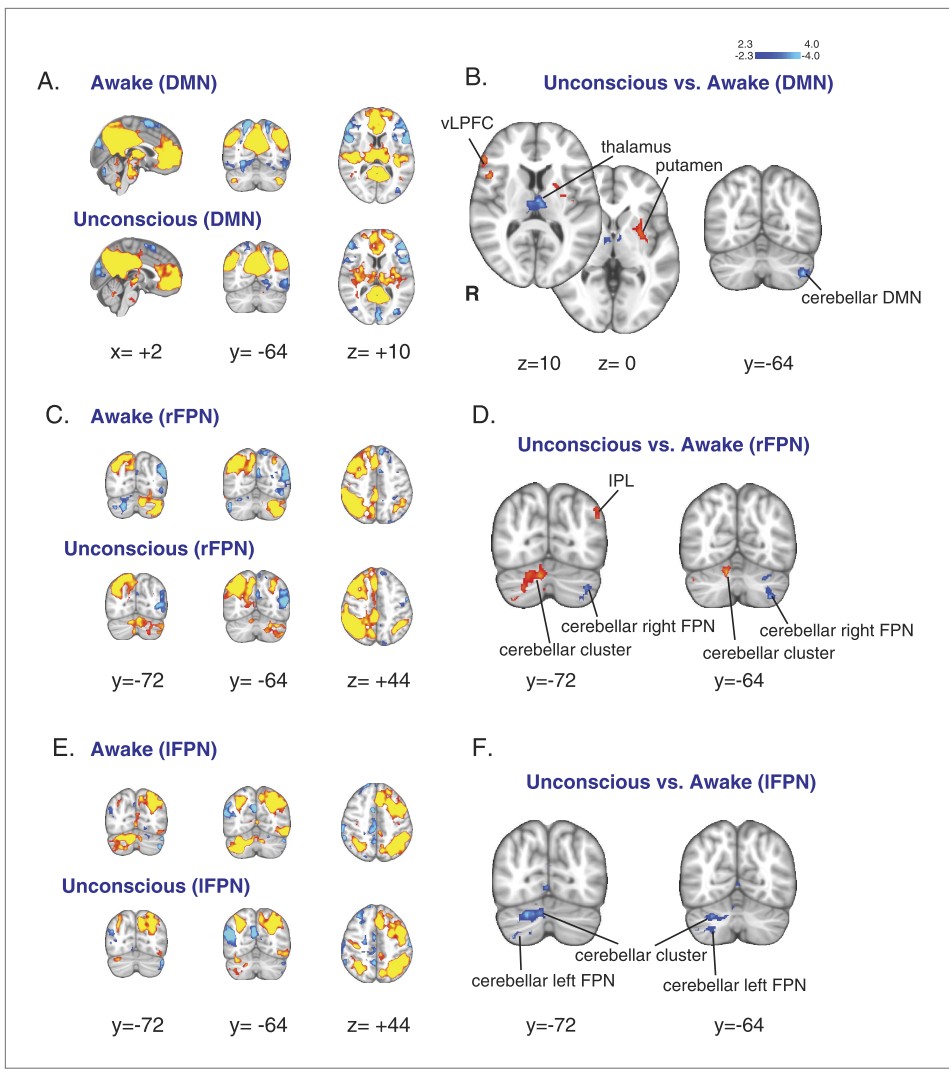

**Figure 3**. Changes in the default mode and the bilateral frontal parietal networks during unconsciousness. (**A**) The DMN extracted from BOLD signals during the awake (n = 16) and unconscious states (n = 16) and displayed on the MNI152 standard volume. (**B**) A comparison the DMN network during the unconscious vs the awake state identified the putamen and the vlPFC as regions that exhibited increased functional connectivity during dexmedetomidine-induced unconsciousness. This comparison also identified the thalamus and the cerebellar representation of this network as regions that exhibited decreased functional connectivity during dexmedetomidine-induced unconsciousness. Cortico–cortico functional connectivity within this network was maintained. (**C**) The right FPN extracted from BOLD signals during the awake and unconscious states (n = 16) and displayed on the MNI152 standard volume. (**D**) A comparison of the right FPN network during the unconscious vs the awake state identified the infraparietal lobule and a right cerebellar cluster as regions that exhibited increased functional connectivity during dexmedetomidine-induced unconsciousness. This comparison also identified the cerebellar representation of this network as a region that exhibited decreased functional connectivity during unconsciousness. Cortico–cortico functional connectivity within this network was maintained. (**E**) The left FPN extracted from BOLD signals during the awake and unconscious states (n = 16) and displayed on the MNI152 standard volume. (**F**) A comparison of the left FPN network during the unconscious vs the awake state identified a right cerebellar cluster and the cerebellar representation of the left FPN as regions that exhibited decreased functional connectivity during dexmedetomidine-induced unconsciousness. Cortico–cortico functional connectivity within this network was maintained. BOLD, blood oxygen level dependent; DMN, default mode network; FPN, Frontoparietal Network; R, right; vlPFC, ventrolateral prefrontal cortex.

The following figure supplement is available for figure 3:

**Figure supplement 1**. SUV, CMR$_{glc}$, rCBF changes during unconsciousness and the relationship to the DMN and bilateral FCNs.

**Table 3.** DMN

| | | MNI coordinate (mm) | | | | |
|---|---|---|---|---|---|---|
| Label | Z-stat | x | y | z | cluster size (# voxels) | cluster p-value |
| awake > unconscious | | | | | | |
| R thalamus | 4.04 | 2 | −14 | 8 | 438 | 0.00193 |
| L thalamus | 3.9 | −4 | −14 | 8 | | |
| L cerebellum (Crus II) | 3.93 | −44 | −64 | −44 | 270 | 0.0429 |
| unconscious > awake | | | | | | |
| R inferior frontal gyrus, pars opercularis | 3.83 | 56 | 12 | 20 | 399 | 0.00381 |
| R inferior frontal gyrus, pars triangularis | 3.35 | 54 | 32 | 10 | | |
| L putamen | 3.47 | −24 | 6 | −8 | 382 | 0.00517 |
| L insula | 3.43 | −36 | −10 | −4 | | |

DMN, Default Mode Network; L, left; MNI, Montreal Neurological Institute; R, right.

connectivity is limited because our methods are not sensitive to small focal differences. We also found that functional connectivity of the left cerebellar representation of the DMN (*Buckner et al., 2011*) was significantly reduced during unconsciousness (*Figure 3B*, *Table 3*). When we compared the FPNs in these two states, we found that the left and right FPNs showed opposite changes in functional connectivity with a cerebellar cluster, likely indicating a switch in hemispheric dominance for cortico-cerebellar functional connectivity to this region during unconsciousness (*Figure 3C–F*, *Table 4*, *Table 5*). Functional connectivity of the cerebellar representations of the FPNs (*Buckner et al., 2011*) was also significantly reduced (*Figure 3D,F*, *Table 4*, *Table 5*). Since cortico-cerebellar connections are mediated by the thalamus (*Guillery and Sherman, 2002*), the changes we observed in cortico-cerebellar functional connectivity are likely related to impaired thalamic functioning. Taken together, these results show that

**Table 4.** rFCN

| | | MNI coordinate (mm) | | | | |
|---|---|---|---|---|---|---|
| Label | Z-stat | x | y | z | cluster size (# voxels) | cluster p-value |
| awake > unconscious | | | | | | |
| L cerebellum (Crus II) | 3.99 | −36 | −70 | −40 | 472 | 0.000724 |
| L cerebellum (VIIb) | 3.83 | −34 | −62 | −50 | | |
| L cerebellum (VI) | 3.28 | −28 | −62 | −32 | | |
| unconscious > awake | | | | | | |
| R cerebellum (VI) | 4.02 | 12 | −68 | −28 | 693 | 1.97E-05 |
| R cerebellum (Crus II) | 3.67 | 26 | −78 | −42 | | |
| R cerebellum (Crus I) | 3.24 | 20 | −74 | −24 | | |
| R precuneus | 2.99 | 26 | −60 | 18 | 519 | 0.000323 |
| R parietal operculum | 4.1 | 58 | −12 | 12 | 516 | 0.00034 |
| R insula | 3.23 | 44 | −4 | 0 | | |
| R cerebellum (Crus I) | 4.28 | 44 | −56 | −32 | 292 | 0.0212 |
| R fusiform gyrus | 3.59 | 30 | −42 | −20 | | |
| L angular gyrus | 3.72 | −38 | −86 | 20 | 272 | 0.0319 |
| L angular gyrus | 3.56 | −48 | −78 | 30 | | |

L, left; MNI, Montreal Neurological Institute; R, right; rFCN, right Frontoparietal Control Network.

**Table 5.** lFCN

| Label | Z-stat | MNI coordinate (mm) | | | cluster size (# voxels) | cluster p-value |
|---|---|---|---|---|---|---|
| | | x | y | z | | |
| awake > unconscious | | | | | | |
| R cerebellum (Crus II) | 4.24 | 38 | −74 | −50 | 1223 | 5.33E-09 |
| R cerebellum (VIIb) | 4.1 | 38 | −56 | −50 | | |
| R cerebellum (VI) | 4.08 | 28 | −62 | −32 | | |
| R intracalcarine cortex | 3.58 | 4 | −86 | −2 | | |
| L middle frontal gyrus | 3.89 | −32 | 12 | 54 | 490 | 0.000341 |
| L superior frontal gyrus, medial part | 3.52 | −4 | 32 | 46 | | |
| unconscious > awake | | | | | | |
| n.s. | | | | | | |

L, left; lFCN, left Frontoparietal Control Network; MNI, Montreal Neurological Institute; n.s, not significant; R, right.

unconsciousness is associated with decreased thalamic CMR$_{glc}$, rCBF, and functional connectivity to the DMN and support the notion that the thalamus plays a critical role in mediating unconsciousness.

## Seed-based thalamic connectivity during unconsciousness showed reduced thalamic functional connectivity to posterior cingulate, precuneus, and inferior parietal cortices

In order to further examine the role of the thalamus in unconsciousness, we next performed a seed-based functional connectivity analysis using the thalamic region with decreased DMN functional connectivity as the seed region. This thalamic seed also overlapped with thalamic regions showing reduced rCBF and CMR$_{glc}$. Our seed-based analysis allowed us to more thoroughly examine changes in thalamic functional connectivity to all brain regions. We found that the thalamic seed was significantly functionally disconnected from posterior regions of the DMN (posterior cingulate cortex, precuneus, and inferior parietal lobules) during unconsciousness (*Figure 4A*, *Table 6*). Notably, the cortical regions showing reduced thalamic functional connectivity also overlapped with those demonstrating reduced flow and metabolism (*Figure 4B*).

## Decreases in rCBF in the DMN, FPN, and thalamus do not immediately reverse during recovery from unconsciousness

We next investigated whether the rCBF changes during the unconscious state returned to baseline during the recovery state. We identified 10 volunteers who responded to verbal instructions (opened eyes and gave a thumbs-up signal, though they were still mildly sedated) during at least 6 min of the recovery ASL scan. Surprisingly, when we compared the recovery state to the unconscious state in these volunteers, we found that the rCBF did not increase (cluster corrected, *Figure 5A*). In fact, when explored at a more liberal threshold (uncorrected, p < 0.05), further decreases in rCBF were observed (*Figure 5B*). When we compared the recovery state to the awake state, we found that the decrease in rCBF showed a spatial distribution similar to that observed during the unconscious state (*Figure 5C*). Next, we aligned all subjects to unconscious and recovery time points in order to study the dynamics of rCBF changes within the statistically significant clusters. We confirmed that rCBF decreased during loss of consciousness and that this decrease was maintained during recovery from this state in all regions analyzed (*Figure 5—figure supplement 1*). These results suggest that cortical rCBF increase was not sufficient for the early stages of recovery from unconsciousness induced by dexmedetomidine.

## Thalamo-cortical and cortical-cerebellar functional connectivity are restored during recovery from dexmedetomidine-induced unconsciousness

We found that thalamic functional connectivity to the DMN was restored during the recovery state (*Figure 6A–B*, *Table 7*). We also found that the opposing left and right FPN changes in the cerebellar cluster (*Figure 3B–C*) were also reversed (*Figure 6C–F*, *Table 7*). When we analyzed the functional

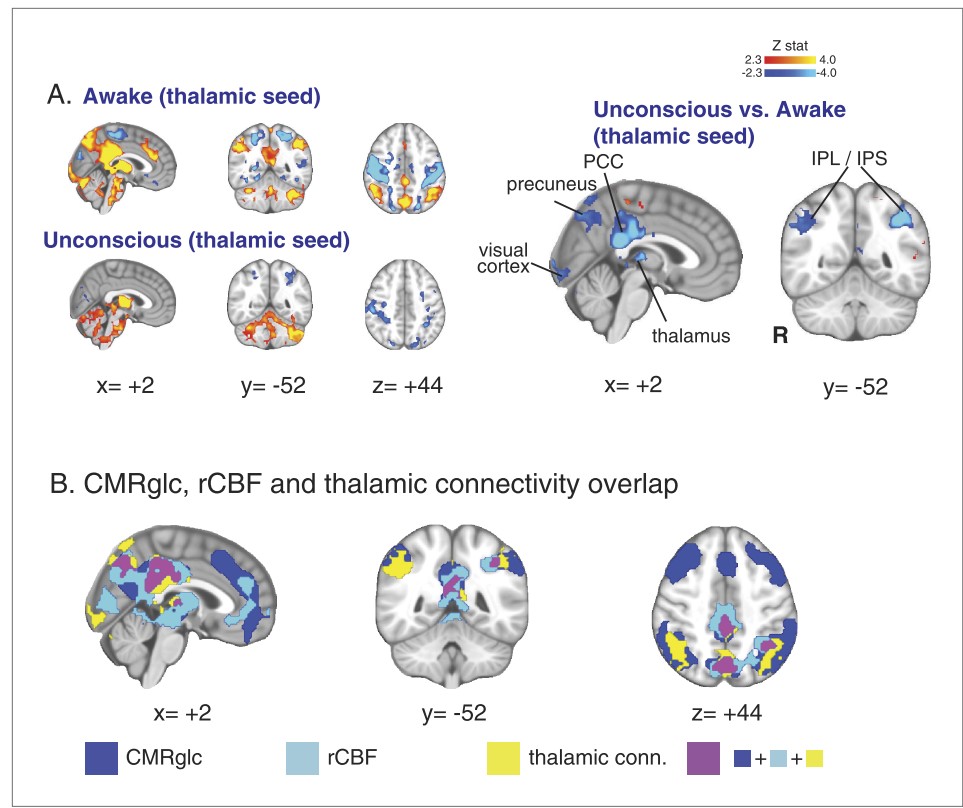

**Figure 4**. Seed based functional connectivity analysis of the brain region with overlapping changes in CMR$_{glc}$, rCBF, and functional connectivity. (**A**) Brain wide representation of regions connected to the thalamic seed during the awake and unconscious states (n = 16). A comparison of thalamic seed functional connectivity during the unconscious vs the awake state identified the PCC, precuneus, thalamus, visual cortex, IPL and IPS as regions exhibiting decreased functional connectivity during unconsciousness. (**B**) Overlap of the brain regions with changes in CMR$_{glc}$, rCBF, and functional connectivity. Notably, the posterior thalamus, PCC and the precuneus exhibited overlapping changes. CMR$_{glc}$, cerebral metabolic rate of glucose; IPL, infraparietal lobule; IPL, infraparietal sulcus; PCC, posterior cingulate cortex; R, right; rCBF, regional cerebral blood flow.

connectivity of our thalamic seed of interest, we found that thalamic functional connectivity to the posterior cingulate cortex and thalamo-thalamic functional connectivity were restored during the recovery state (*Figure 6G–H*, *Table 7*). Taken together, these results provide further evidence that dexmedetomidine-induced unconsciousness is associated with decreased thalamic functional connectivity to the DMN and supports the notion that the thalamus plays a critical role in mediating unconsciousness and recovery from this state.

## Discussion

Resting-state functional brain imaging is increasingly being utilized to probe the mechanisms of consciousness (*Fox and Greicius, 2010*). By using a very site-specific drug, performing for the first time simultaneous measurements of CMR$_{glc}$, rCBF, resting state BOLD low frequency oscillations, and analyzing both dexmedetomidine-induced loss of consciousness and recovery, we have been able to identify a minimal disruption of neural networks that is shared between altered consciousness states. Our results show that during dexmedetomidine-induced unconsciousness, cortico-cortical functional connectivity remains intact but thalamo-cortical functional connectivity is disrupted. This disrupted thalamo-cortical functional connectivity appropriately reverses during recovery from dexmedetomidine-induced unconsciousness.

### Thalamo-cortical correlates of altered arousal and unconsciousness

The notion that disruption of thalamo-cortical functional connectivity could constitute a neural-network mechanism responsible for unconsciousness that is distinct from disruption of cortico-cortical

**Table 6.** Thalamic seed

| Label | Z-stat | MNI coordinate (mm) | | | cluster size (# voxels) | cluster p-value |
|---|---|---|---|---|---|---|
| | | x | y | z | | |
| awake > unconscious | | | | | | |
| R thalamus | 5.33 | 12 | −12 | 8 | 3445 | 4.05E-16 |
| L posterior cingulate cortex | 4.95 | −6 | −36 | 30 | | |
| R posterior cingulate cortex | 4.73 | 6 | −28 | 34 | | |
| L thalamus | 4.48 | −16 | −10 | 10 | | |
| R caudate nucleus | 3.44 | 12 | 10 | 18 | | |
| L globus pallidus | 3.44 | −12 | 2 | −2 | | |
| L precuneus | 4.03 | −6 | −56 | 36 | 1230 | 1.79E-07 |
| R precuneus | 3.71 | 12 | −60 | 36 | | |
| R angular gyrus | 4.07 | 36 | −60 | 50 | 1094 | 9.54E-07 |
| R supramarginal gyrus | 3.67 | 48 | −48 | 36 | | |
| R cerebellum (crus II) | 4.07 | 8 | −76 | −30 | 906 | 8.76E-06 |
| R cerebellum (crus I) | 3.84 | 22 | −88 | −22 | | |
| R intracalcarine cortex | 3.75 | 2 | −100 | 4 | | |
| L cerebellum (crus I) | 3.14 | −20 | −84 | −28 | | |
| L supramarginal gyrus | 4.6 | −38 | −52 | 42 | 831 | 2.21E-05 |
| L angular gyrus | 3.43 | −32 | −76 | 48 | | |
| unconscious > awake | | | | | | |
| L inferior frontal gyrus, pars opercularis | 4.21 | −56 | 14 | 8 | 1458 | 1.80E-08 |
| L superior temporal gyrus | 4.2 | −62 | −10 | 4 | | |
| L middle temporal gyrus | 4.04 | −50 | −2 | −22 | | |
| L insula | 3.55 | −38 | 4 | −6 | | |
| R paracentral lobule | 3.77 | 4 | −32 | 58 | 963 | 4.41E-06 |
| L paracentral lobule | 3.64 | −10 | −38 | 58 | | |
| R superior frontal gyrus, lateral part | 3.44 | 14 | −6 | 56 | | |
| R precentral gyrus | 3.23 | 14 | −20 | 66 | | |
| L postcentral gyrus | 3.2 | −16 | −44 | 72 | | |
| R superior frontal gyrus, medial part | 2.86 | 10 | 2 | 50 | | |
| R postcentral gyrus | 2.72 | 16 | −38 | 70 | | |
| R superior temporal gyrus | 5 | 48 | −42 | 10 | 868 | 1.39E-05 |
| R middle temporal gyrus | 4.53 | 44 | −28 | −6 | | |
| R precentral gyrus | 3.51 | 54 | −2 | 46 | | |
| L postcentral gyrus | 4.03 | −62 | −8 | 36 | 453 | 0.00401 |
| L precentral gyrus | 3.77 | −60 | −4 | 36 | | |
| L superior temporal gyrus | 3.97 | −66 | −44 | 10 | 340 | 0.0244 |

L, left; MNI, Montreal Neurological Institute; R, right.

functional connectivity is supported by a number of findings. During, NREM I and II sleep in humans, DMN cortico-cortical functional connectivity remains preserved (*Horovitz et al., 2008*; *Larson-Prior et al., 2009*) while thalamo-cortical functional connectivity is disrupted (*Picchioni et al., 2014*). However, during NREM III, there is a functional disconnection between frontal and parietal nodes of the DMN (*Horovitz et al., 2009*; *Samann et al., 2011*), and disrupted thalamo-cortical functional connectivity (*Picchioni et al., 2014*). Likewise, in-depth electrode studies in humans show that changes

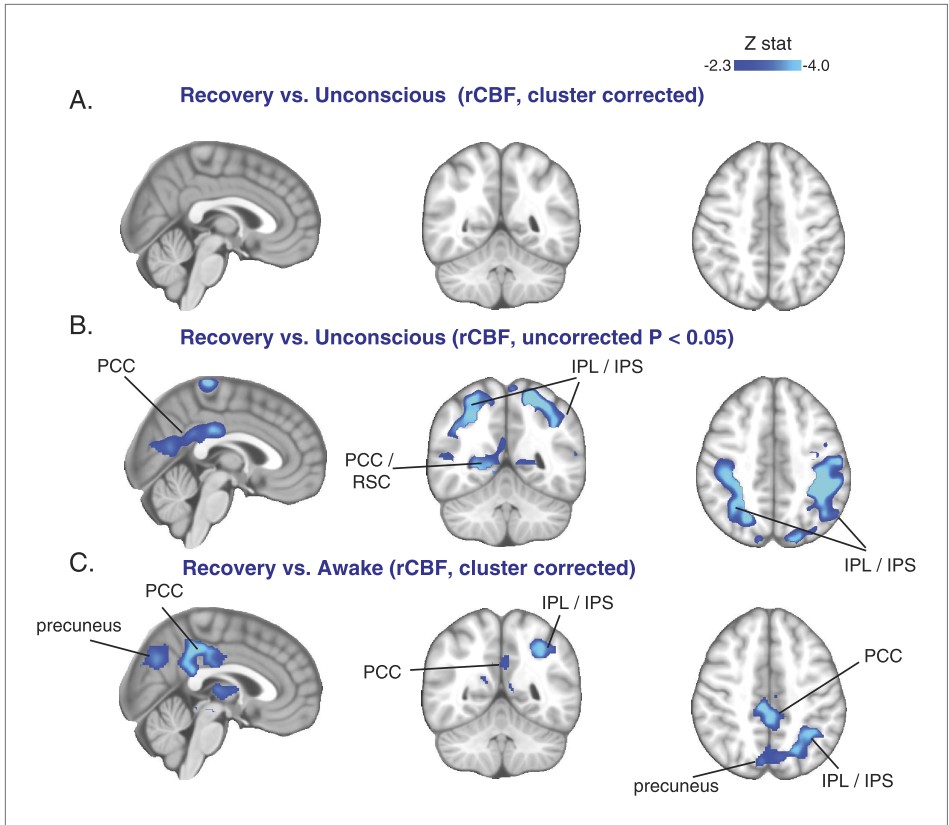

**Figure 5**. Cortical increase in rCBF is not evident during the recovery of consciousness. (**A**) Cluster corrected (n = 10) rCBF comparison of the recovery state vs the unconsciousness state displayed on the MNI152 standard volume highlighting no change in rCBF at recovery. (**B**) Uncorrected (n = 10) rCBF comparison of the recovery state vs the unconsciousness state displayed on the MNI152 standard volume suggesting that a decrease in rCBF occurred during the recovery state. (**C**) Cluster corrected (n = 10) rCBF comparison of the recovery state vs the awake state displayed on the MNI152 standard volume highlighting a spatial distribution of rCBF decrease similar to that observed during unconscious vs the awake state (n = 17) comparison. IPL, infraparietal lobule; IPL, infraparietal sulcus; PCC, posterior cingulate cortex; rCBF, regional cerebral blood flow; R, right; RSC, retrosplenial cortex. The brain coverage of rCBF estimation is shown in *Figure 1D*.

The following figure supplement is available for figure 5:

**Figure supplement 1**. Cortical decrease in rCBF persists during the recovery of consciousness.

---

in thalamic activity during sleep precedes changes in cortical activity, suggesting that disrupted thalamo-cortical functional connectivity may underlie changes in consciousness (*Magnin et al., 2010*). Also, during propofol anesthesia-induced unconsciousness, decreases in thalamo-cortical functional connectivity, alongside disruptions in cortico-cortical functional connectivity in the DMN and FPNs, have been described in healthy volunteers (*Boveroux et al., 2010*; *Ni Mhuircheartaigh et al., 2013*) Functional reintegration of the thalamus to key regions of the cortex appears to be necessary for the recovery of consciousness from this state (*Langsjo et al., 2012*).

## A putative functional network for recovery from unconsciousness comprising the locus ceruleus, central thalamus, and posterior cingulate cortex

Norepinephrinergic neurons in the locus ceruleus project to thalamic mediodorsal, midline, intralaminar nuclei, which in turn have neurons that project to the cingulate cortex (*Jones and Yang, 1985*; *Vogt et al., 1987*; *Buckwalter et al., 2008*; *Vogt et al., 2008*). Our results show that these connections may comprise a functional circuit that is disrupted during dexmedetomidine-induced unconsciousness. Moreover, this functional circuit appears specifically to involve the posterior cingulate

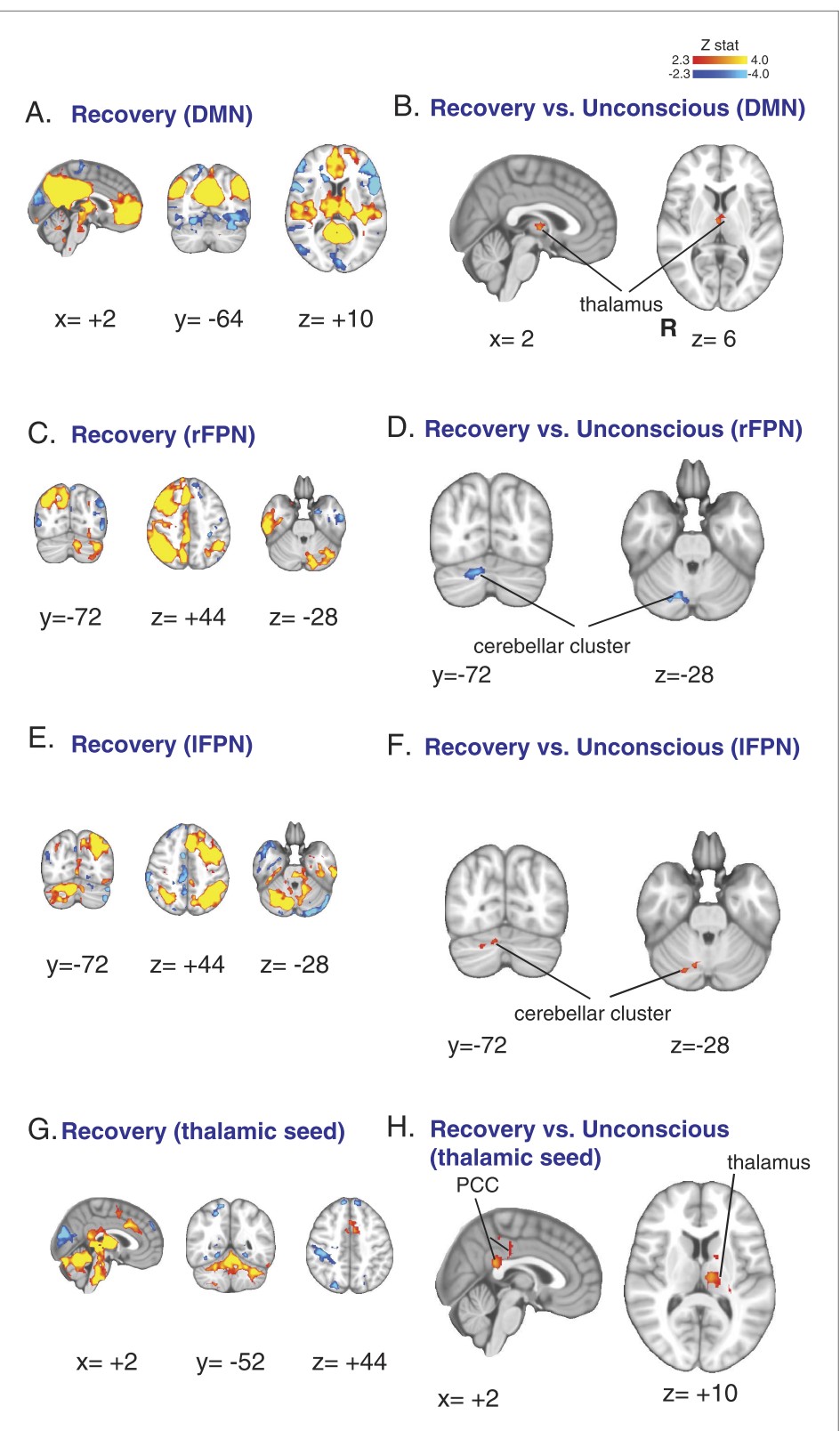

**Figure 6**. Functional connectivity changes observed during recovery of consciousness. (**A**) The DMN extracted from BOLD signals obtained during the recovery state (n = 15) displayed on the MNI152 standard volume. (**B**) A comparison the DMN network during the recovery vs the awake state identified the thalamus as the only

*Figure 6. Continued on next page*

*Figure 6. Continued*

region with partial recovery of functional connectivity. (**C**) The right FPN extracted from BOLD signals during the recovery state (n = 15) and displayed on the MNI152 standard volume. (**D**) A comparison of the right FPN during the recovery vs the awake state showed that right cerebellar cluster from *Figure 3B,C* now exhibited decreased right FPN functional connectivity. (**E**) The left FPN extracted from BOLD signals during the recovery state (n = 15) and displayed on the MNI152 standard volume. (**F**) A comparison of the left FPN during the recovery vs the awake state showed that right cerebellar cluster from *Figure 3B,C* now exhibited increased left FPN functional connectivity. (**G**) Brain wide representation of regions connected to the thalamic seed during the recovery state (n = 15) displayed on the MNI152 standard volume. (**H**) A comparison of thalamic seed functional connectivity during the recovery vs the awake state identified the PCC and thalamus as regions with increased functional connectivity at recovery. BOLD, blood oxygen level dependent; DMN, default mode network; FPN, Frontoparietal Network; PCC, posterior cingulate cortex; R, right.

cortex, distinct from anterior or mid-cingulate cortex. In our thalamic seed-based functional connectivity analysis, we found that during dexmedetomidine-induced unconsciousness, thalamic regions consistent with intralaminar, midline, mediodorsal, and ventral anterior nuclei were functionally disconnected from posterior DMN structures (precuneus and the posterior cingulate cortex). Perhaps more importantly, recovery from dexmedetomidine-induced unconsciousness was associated with restored thalamo-thalamic and thalamic functional connectivity to the posterior cingulate cortex. Structural connectivity between the thalamus and the posterior cingulate cortex has also recently been correlated with the severity of DOC and used to subcategorize patients in minimally conscious states (*Fernandez-Espejo et al., 2012*). The above observations lead us to conclude that a noradrenergic circuit involving the thalamus and posterior cingulate cortex may be important for recovery from both dexmedetomidine-induced unconsciousness and DOC states.

## Reduced cerebral blood flow during recovery of consciousness

An FDG-PET study has recently described a linear relationship between metabolism in the precuneus and central thalamus in severely brain-injured patients with DOC (*Fridman et al., 2014*). We note that our observed reduction in blood flow in the precuneus and thalamus during dexmedetomidine-induced unconsciousness parallels this result. However, during recovery from dexmedetomidine-induced unconsciousness, we did not observe an increase in rCBF to any brain regions. Since our current and previously reported (*Drummond et al., 2008*) results suggest that dexmedetomidine does not decouple rCBF and CMRglc, we surmise that metabolism does not return to baseline during the earliest phases of recovery from dexmedetomidine-induced unconsciousness. Thus, the reduced rCBF we report after recovery from dexmedetomidine-induced unconsciousness could reflect a residual drug effect state in which subjects are responsive to external stimuli but remain in a sedated state. This residual effect may be mediated by the broad and diffuse nor-adrenergic projections from the locus ceruleus to the rest of the brain (*Vogt et al., 2008*).

This result is interesting because it suggests that recovery from unconsciousness may occur in a graded fashion. For example, performance on more complex cognitive tasks likely differs substantially between the awake and recovery states. This suggests that re-establishment of functional connectivity between vital brain regions precedes increases in rCBF and CMRglc. Return of rCBF and CMRglc to baseline levels may be indicative of the final stages of recovery from dexmedetomidine-induced unconsciousness. Future investigations incorporating cognitive and psychomotor vigilance tasks during recovery from experimental models of unconsciousness would lend further insight on whether this is a drug-specific effect. Nevertheless, this finding is interesting because it suggests that recovery from unconsciousness states may proceed prior to significant increases in rCBF.

## Different states of unconsciousness may reflect a hierarchy of disruption in functional circuits

Drawing from previous work on sleep (*Horovitz et al., 2008, 2009; Larson-Prior et al., 2009; Magnin et al., 2010; Samann et al., 2011; Picchioni et al., 2014*) and anesthesia (*White and Alkire, 2003; Alkire et al., 2008; Boveroux et al., 2010; Liu et al., 2013; Ni Mhuircheartaigh et al., 2013*), our results suggest a hierarchy of brain network disruptions that can result in different states of altered consciousness. In NREM II (non-slow wave) sleep and dexmedetomidine-induced unconsciousness,

**Table 7.** Recovery

| Label | Z-stat | MNI coordinate (mm) | | | cluster size (# voxels) | cluster p-value |
|---|---|---|---|---|---|---|
| | | x | y | z | | |
| DMN, recovery > unconscious | | | | | | |
| R thalamus | 3.3 | 2 | −8 | 4 | 78 | 0.0345 |
| L thalamus | 3 | −4 | −4 | 8 | | |
| DMN, unconscious > recovery | | | | | | |
| n.s. | | | | | | |
| rFCN, recovery > unconscious | | | | | | |
| n.s. | | | | | | |
| rFCN, unconscious > recovery | | | | | | |
| R cerebellum (VI) | 3.92 | 14 | −70 | −28 | 246 | 0.00172 |
| R cerebellum (Crus I) | 3.81 | 22 | −76 | −28 | | |
| R cerebellum (Crus II) | 3.06 | 24 | −80 | −38 | | |
| lFCN, recovery > unconscious | | | | | | |
| R cerebellum (VI) | 3.58 | 12 | −66 | −30 | 118 | 0.012 |
| R cerebellum (Crus I) | 3.29 | 38 | −58 | −32 | 93 | 0.0232 |
| R cerebellum (Crus II) | 3.05 | 32 | −64 | −42 | | |
| lFCN, unconscious > recovery | | | | | | |
| n.s. | | | | | | |
| Thalamic seed, recovery > unconscious | | | | | | |
| posterior cingulate cortex | 3.53 | 0 | −40 | 22 | 367 | 0.00367 |
| L thalamus | 3.49 | −10 | −22 | 12 | 348 | 0.00471 |
| L putamen | 2.51 | −14 | 4 | 2 | | |
| Thalamic seed, unconscious > recovery | | | | | | |
| n.s. | | | | | | |

DMN, default mode network; L, left; lFCN, left Frontoparietal Control Network; MNI, Montreal Neurological Institute; n.s, not significant; R, right; rFCN, right Frontoparietal Control Network.

patients can be easily aroused to respond to verbal commands by sufficiently strong external stimuli. In these states, cortico-cortical functional connectivity is preserved (*Horovitz et al., 2008*; *Larson-Prior et al., 2009*), but thalamo-cortical functional connectivity is disrupted (*Picchioni et al., 2014*). In propofol anesthesia-induced unconsciousness and some DOC states, where patients cannot be aroused by external stimuli, both cortico-cortical and thalamo-cortical functional connectivity are disrupted (*Boveroux et al., 2010*).

In the case of sleep, dexmedetomidine-induced unconsciousness, and some minimally conscious states, maintained cortico-cortical functional connectivity likely allows for the cortex to be 'primed', ready to recover from the altered consciousness states with restoration of thalamic functional connectivity, which could be triggered by ascending midbrain and brainstem inputs (*Solt et al., 2014*) or direct therapeutic intervention at the thalamic level (*Schiff et al., 2007*; *Williams et al., 2013*). In other altered consciousness states, such as during general anesthesia or comatose states, markedly disrupted cortico-cortical functional connectivity could further impair the ability to recover consciousness. We note that while it might be theoretically possible to attain unconsciousness by disrupting only cortico-cortical functional connectivity while preserving thalamo-cortical functional connectivity, it appears, from our study and others, that this does not happen during sleep and anesthesia-induced unconsciousness.

Compared to the findings supporting a role for disrupted cortico-cortical functional connectivity during unconsciousness, our findings suggest that thalamo-cortical connectivity constitutes a more

fundamental disruption required to produce unconsciousness. This insight could lead to a more objective characterization, diagnosis, and prognosis of DOC. Also, the state of cortico-cortical and thalamo-cortical functional connectivity could be used to better select patients who may benefit from therapeutic interventions. For instance, in the event of maintained cortico-cortical functional connectivity but impaired thalamo-cortical functional connectivity with intact thalamo-cortical structural connections (*Fernandez-Espejo et al., 2012*), therapies aimed at restoring thalamo-cortical functional connectivity might be more likely to help patients recover. These therapies could be pharmacological (*Williams et al., 2013*), electrical (*Schiff et al., 2007*), or could involve other forms of stimulation. Since dexmedetomidine appears to disrupt functional network activity in a manner similar to NREM II sleep, our results also suggest that alpha-2-receptor agonists could be further developed as effective sleep therapeutic drugs.

## Subjective awareness during sleep and anesthesia

At present, there is a debate pertaining to whether anesthetized and unresponsive patients (without motor impairments) can remain internally conscious to subjective experiences (internally generated or provoked by external stimuli). This debate has been fueled by clinical studies of anesthetized patients that have reported rare intraoperative episodes of awareness (resulting from light anesthetic depth) under anesthesia (*Ghoneim et al., 2009*) and dreaming under anesthesia (*Brandner et al., 1997*; *Leslie et al., 2005*; *Errando et al., 2008*; *Sanders et al., 2012*). Noreika et al. studied this phenomenon with dexmedetomidine, sevoflurane, propofol, and xenon (*Noreika et al., 2011*) and confirmed that subjective experiences of consciousness occur during clinically defined unconsciousness states. However, the experimental paradigm that was used to test for subjective experiences of consciousness involved a gradual increase in drug concentrations. This empiric gradual increase did not directly target anesthetic drug dosing to a neurophysiologically defined brain state nor does it inform us on the amount of time that was spent in lighter levels of anesthesia. Thus, the reported subjective experiences could have occurred when the drug concentration levels were very low. This line of reasoning is supported by evidence confirming that subjective experiences of dreaming under general anesthesia likely occur when patients are emerging from general anesthesia (*Leslie et al., 2009*).

Subjective experiences of consciousness also occur during REM sleep (*Hobson, 2009*). To a limited degree, they are also thought to occur during NREM sleep, especially at sleep onset in NREM I and during NREM II sleep periods encountered later at night (*Hobson, 2009*). However, the current methods for scoring sleep states remain imprecise and highly subjective, with limited state and temporal resolution, and high intra- and inter-scorer variability, raising the possibility that clinically scored NREM I and II sleep may include periods of arousal. In clinical sleep scoring, trained technicians visually score the sleep time-series data in 30-s epochs according to semantically defined sleep stages. These scoring standards grossly simplify sleep electroencephalographic dynamics. For example, consider the characterization of sleep onset. The American Academy of Sleep Medicine defines sleep onset as the first appearance of any 30-s epoch that contains at least 15 s of sleep (*Iber et al., 2007*). This implies that sleep onset is a binary process. However, recent EEG studies in humans clearly demonstrate that sleep onset, at both a neurophysiological and behavioral level, is a continuous process. As such, methods designed to track moment-to-moment continuous variations in the EEG are significantly better at predicting when subjects lose wakefulness, measured in terms of a behavioral task, than traditional sleep scoring methods (*Prerau et al., 2014*). In particular, subjects can continue to perform awake behavior despite being scored as asleep (NREM I) using traditional sleep scoring methods. Thus, it is possible that the subjective experiences of consciousness that have been ascribed to NREM sleep could have occurred during subjectively scored sleep stages that included un-scored periods of arousal or wakefulness.

Despite this rapidly evolving understanding of subjective experiences during altered arousal under sleep and anesthesia, we have tried in this study to focus on a particular state of dexmedetomidine-induced unconsciousness. We achieved this state through rapid administration of dexmedetomidine and confirmed that this dose and form of drug administration produces a NREM II-like state (*Akeju et al., 2014*). We then applied a combination of fMRI and FDG-PET imaging that would characterize the overall time-integrated brain activity during this state. Clinical evidence shows that motor-evoked potentials, and thus motor function from cortex down, remain intact in this state, making it unlikely that the loss of responsiveness we observed is only a result of disrupted motor function (*Moore et al., 2006*;

*Garavaglia et al., 2014*). Therefore, we feel it is appropriate to refer to the state of altered arousal described in this manuscript as dexmedetomdine-induced unconsciousness.

A long-standing debate in anesthesiology relates to the fundamental systems-level mechanisms for anesthesia-induced unconsciousness, with some investigators proposing a thalamic switch (*Alkire et al., 2000*) and others proposing disrupted cortico-cortical connections as the central mechanism (*Massimini et al., 2005*; *Mashour, 2006*; *Alkire et al., 2008*). Our results, in combination with previous work, resolve this conflict by showing how both mechanisms could co-exist but at different parts of the continuum of altered arousal or unconsciousness. Moreover, it suggests a mechanistic interpretation for varying levels of anesthesia-induced unconsciousness, where disrupted thalamo-cortical functional connectivity, but intact cortico-cortical functional connectivity reflects lighter states of unconsciousness and disruptions of both cortico-cortical and thalamo-cortical functional connectivity reflect deeper states of unconsciousness. These insights could guide development of system-specific anesthetic drugs that possess minimal side effects and improved monitoring of patients in both the operating room and intensive care settings.

## Materials and methods

### Subject selection

This study was conducted at the Athinoula A Martinos Center for Biomedical Imaging at the Massachusetts General Hospital. The Human Research Committee and the Radioactive Drug Research Committee at the Massachusetts General Hospital approved the study protocol. After an initial email/ phone screen, potential study subjects were invited to participate in a screening visit. At the screening visit, informed consent including the consent to publish was requested after the nature and possible consequences of the study was explained. All subjects provided informed consent and were American Society of Anesthesiology Physical Status I with Mallampati Class I airway anatomy. After providing informed consent, a standard pre-anesthesia assessment was administered and a blood toxicology screen was performed to ensure that subjects were not taking drugs that might confound the interpretation of study results. A complete metabolic panel and a urine pregnancy test were also obtained to confirm that all values were within normal ranges and to confirm non-pregnant status, respectively. A total of 20 subjects participated in screening visits and 18 subjects were deemed eligible to participate in the study protocol. Two subjects were deemed ineligible after medical evaluation and one subject was lost to follow-up after providing informed consent. During all study visits, subjects were required to take nothing by mouth for at least 8 hr. A urine toxicology screen was performed to ensure that subjects had not taken drugs that might confound interpretation of the results. A pregnancy test was also administered (serum for PET visits, urine for fMRI only visits) for each female subject to confirm non-pregnant status.

### Imaging visit

We studied awake, unconscious, and recovery states in humans, using an integrated positron PET/MR approach, in healthy volunteers, 18–36 years of age. Brain imaging was performed with the Biograph mMR scanner (Siemens Healthcare, Erlangen, Germany), which allows simultaneous acquisition of whole-body PET and 3 Tesla MRI data. The fully integrated PET detectors use avalanche photoiodide technology and lutetium oxyorthosilicate crystals (8 × 8 arrays of $4 \times 4 \times 20$ mm$^3$ crystals). The PET scanner's transaxial and axial fields of view are 594 mm and 25.8 cm, respectively (*Drzezga et al., 2012*). Approximately, 5 mCi of FDG was purchased from an outside approved vendor and was administered as a bolus immediately before the initiation of the dexmedetomidine (described below). A PET-compatible 16-channel head and neck array was used to acquire the MR data.

At the beginning of the imaging visit, we acquired structural MRI (MPRAGE volume, TR/TE = 2100/3.24 ms, flip angle = 7°, voxel size = 1 mm isotropic) for the purpose of anatomical localization, spatial normalization, and surface visualization of the imaging data. A 6:08 min pASL scan and a 6:15 min BOLD fMRI scans were performed ('awake' pASL and BOLD scans; *Figure 1C*), in order to assess rCBF and functional connectivity at baseline. The pASL scans were collected using the 'PICORE-Q2TIPS' MRI labeling method (*Luh et al., 1999*) (TR/TE/TI1/TI2 = 3000/13/700/1700 ms, voxel size = 3.5*3.5*5 mm, number of slices = 16). 'Tag' images were acquired by labeling a thick inversion slab (110 mm), proximal to the imaging slices (gap = 21.1 mm). 'Control' images were acquired interleaved with the tag images, by applying an off-resonance inversion pulse without any spatial

encoding gradient. At the beginning of each pASL scan, an $M_0$ scan (i.e., the longitudinal magnetization of fully relaxed tissue) was acquired for rCBF quantification purposes. The pASL imaging volume common to all subjects scanned covered most of the cerebrum, and extended ventrally to the midbrain and dorsally to the vertex (*Figure 1D*). BOLD fMRI data were collected using a whole brain T2*-weighted gradient echo BOLD echo planar imaging pulse sequence was used (TR/TE = 3000/35 ms, flip angle = 90°, voxel size = 2.3 × 2.3 × 3.8 mm, number of slices = 35).

After the awake scans, dexmedetomidine was administered as a 1 mcg/kg loading bolus over 10 min, followed by a 0.7 mcg/kg/hr infusion to maintain unconsciousness. Another pASL scan was initiated at the onset of the dexmedetomidine infusion, with identical imaging parameters as the awake pASL scan, except for a longer duration (20:08 min). This longer acquisition was performed in order to ensure that perfusion data were collected for a long enough period to capture the transition from the awake to the unconscious state. During the infusion period, the study anesthesiologists monitored cuff blood pressure, capnography, electrocardiogram, and pulse-oximetry. Volunteers were instructed to keep their eyes open during the course of the study; loss of consciousness was defined as the onset of sustained eye closure and lack of response to a verbal request to open the eyes. After the onset of unconsciousness, we then performed a 6:15 min 'unconscious' BOLD fMRI scan (to evaluate unconsciousness-related changes in functional connectivity). After acquisition of all images, the dexmedetomidine infusion was discontinued, and a 20:08 min pASL scan was performed to assess changes in rCBF upon recovery. Spontaneous eye opening and a positive response to give a thumbs-up signal were used to determine recovery of consciousness. All 10 subjects who experienced spontaneous eye opening during the final 6:08 min of the recovery of consciousness pASL successfully executed on the thumbs-up signal request. An additional 6:15 min 'recovery' BOLD fMRI scan was performed, in order to assess variations in functional connectivity upon recovery.

## Data analysis

PET data collected from 10 subjects (two visits each) and stored in list mode format were binned into sinograms. In addition to the static frame corresponding to 40–60 min post FDG administration, dynamic frames of progressively longer duration were generated for quantitative analysis. PET images were reconstructed using an ordered subsets expectation maximization algorithm, with 3 iterations and 21 subsets. Corrections were applied to account for variable detector efficiencies and dead time, photon attenuation and scatter and radioactive decay using the software provided by the manufacturer. We employed a method that is similar to the generation of attenuation maps for computed tomography data to generate our head attenuation maps from the MPRAGE data. This head attenuation map was combined with the hardware (i.e., RF coil, patient table, etc) attenuation map provided by the manufacturer. Spatial smoothing was performed post-reconstruction using a 4 mm full-width-at-half-maximum (FWHM) Gaussian kernel.

Standardized uptake values (SUV; i.e., mean radioactivity/injected dose/weight) were computed voxelwise from the emission data collected 40–60 min post-injection. In order to quantitatively assess metabolic changes, cerebral metabolic rate of glucose (CMRglc) were computed from the dynamic PET frames with the non-invasive method proposed by *Wu (2008)*, using the whole-brain as our reference region. SUV and CMRglc maps were then coregistered with the high resolution MRI scan using BBREGISTER tool (*Greve and Fischl, 2009*) from the FreeSurfer suite (http://surfer.nmr.mgh.harvard.edu/), normalized to the Montreal Neurological Institute (MNI) space using nonlinear registration (FNIRT, from the FSL suite; FMRIB's Software Library, www.fmrib.ox.ac.uk/fsl/) (*Smith et al., 2004*) and then smoothed with an 8 mm FWHM kernel. A paired, whole-brain, voxel wise, mixed effects analysis was conducted to compare SUV and CMRglc across visits (n = 10). Statistical parametric maps were thresholded using clusters determined by a voxel-wise threshold (Z > 2.3) and a (corrected) cluster significance threshold of p = 0.05 (*Worsley, 2001*).

fMRI data were collected from 17 subjects. However, BOLD data (awake, unconscious, and recovery) were successfully collected in only 15 subjects. This is because BOLD data was collected with a different set of parameters in one subject and another subject exited the scanner immediately prior to the recovery BOLD scan. ASL data preprocessing was performed using a combination of analysis packages including FSL, and Freesurfer. The 'tag', 'control', and $M_0$ scans were first motion-corrected using MCFLIRT (*Jenkinson et al., 2002*). Then, tag and control scans were surround subtracted (i.e., given each $tag_X$, $[(control_{X-1} + control_{X+1})/2 - tag_X]$) to achieve perfusion-weighted images. Quantification of rCBF was performed using the ASLtbx toolbox (*Wang et al., 2008*; *Wang, 2012*)

from the following data: the whole 6 min awake scan, the last 6 min of the 20 min induction of unconsciousness scan (i.e., the portion of the scan during which all subjects were unconscious; 'unconscious' scan) and the last 6 min of the 20-min recovery of consciousness scan (i.e., the portion of the scan during which 10 subjects had recovered; 'recovery' scan).

rCBF maps were coregistered with the high resolution MR scan using Freesurfer's BBREGISTER tool (*Greve and Fischl, 2009*), normalized to MNI space using nonlinear registration (FNIRT) (*Smith et al., 2004*) and then smoothed with a 7 mm FWHM kernel. A paired, voxel wise, mixed effects analyses were conducted to compare rCBF between the awake and unconscious states data (n = 17). This contrast was computed only within voxels imaged in all subjects (see *Figure 2D* for coverage common to all subjects), as assessed through the computation of a conjunction mask from all the MNI-registered $M_0$ scans. Given that rCBF estimation in the white matter presents methodological challenges (particularly given the poor signal-to-noise ratio and longer arterial transit time) (*Liu et al., 2010*), voxels classified as white matter in (the arbitrary value of) ≥ 70% of subjects by the Harvard–Oxford Subcortical Structural Atlas (Center for Morphometric Analyses, http://www.cma.mgh.harvard.edu/fsl_atlas.html) were excluded from the analyses.

Subsequently, another paired analysis was performed to compare rCBF between the unconscious and recovery states. In this analysis, only the 10 subjects who were awake for the full duration of the last 6 min of the 20-min recovery scans were included. This analysis was focused within search areas determined by the statistically significant clusters in the 'unconscious vs awake' contrast. Unless otherwise specified, all rCBF statistical parametric maps were thresholded using clusters determined by a voxel-wise threshold (Z > 2.3) and a (corrected) cluster significance threshold of p = 0.05 (*Worsley, 2001*).

Resting-state BOLD fMRI data were used to perform functional connectivity analyses. Datasets were slice-timing corrected (SLICETIMER), motion-corrected (MCFLIRT), brain-extracted (BET), registered to MPRAGE (BBREGISTER) and then to the MNI152 atlas (FNIRT), smoothed with a 5 mm FMWH kernel and high-pass filtered (cutoff = 0.008 Hz). As ASL and PET analyses revealed that unconsciousness was associated with a reduction in glucose metabolism and rCBF in regions belonging to distinct resting-state networks (Default Mode and both Frontoparietal Control networks; DMN, FCN), we assessed how intrinsic functional connectivity of these specific networks was affected by unconsciousness, using the dual regression approach (*Filippini et al., 2009*; *Zuo et al., 2010*). All the preprocessed datasets were concatenated to create a single 4D dataset. A probabilistic Independent Component Analysis (pICA) (*Beckmann et al., 2005*) was performed using Multivariate Exploratory Linear Optimized Decomposition into Independent Components (MELODIC) on this concatenated 4D dataset, limiting the number of independent components (ICs) to 25, as in previous publications using the dual regression approach (*Filippini et al., 2009*; *Napadow et al., 2010*, *2012*; *Kim et al., 2013*; *Loggia et al., 2013*). From the pool of the 25 ICs, the DMN, right and left FCNs were clearly identified. Group-level spatial maps for each of these RSNs were used as a set of spatial regressors in a General Linear Model (GLM), in order to identify the individual subjects' time course associated with each group-level map. These time courses were then variance normalized and used as a set of temporal regressors in a GLM, to find subject-specific maps associated with the different group-level independent components. In this GLM, explanatory variables also included six motion parameters and the time courses from ventricles and white matter as covariates of no interest. Subject-specific maps were compared across states (awake, unconscious, recovery) using paired t-tests. As our dual regression ICA approach identified altered DMN functional connectivity to the thalamus, we used this thalamic region as a seed for seed-based functional connectivity analysis. This follow-up analysis was done with the purpose of (1) determining whether specific portions of the DMN (e.g., more posterior regions within the network) may be responsible for driving the effect and (2) assessing changes in functional connectivity between the thalamus and other, non-DMN regions. The extracted fMRI time series, variance normalized, was used as a regressor in a GLM for the awake, unconscious, and recovery data. The same nuisance regressors adopted in the dual regression analyses (see above), with the addition of global signal time course were used in the seed-based analyses. Statistical parametric maps were thresholded using clusters determined by a voxel-wise threshold (Z > 2.3) and a (corrected) cluster significance threshold of p = 0.05 (*Worsley, 2001*). From representative clusters identified in the various analyses, data were extracted and displayed for illustrative purposes. Human brain atlases were used for anatomical reference of the cerebrum (*Mai et al., 2008*) and the cerebellum (*Diedrichsen et al., 2009*).

## Acknowledgements

We thank P McCarthy, A Omoruan, K Shelton, and Norman Taylor for providing clinical care; J Kim for the acquisition of physiological data. We also thank the Massachusetts General Hospital Clinical Research Center for providing nursing support.

## Additional information

### Funding

| Funder | Grant reference number | Author |
| --- | --- | --- |
| Foundation for Anesthesia Education and Research | | Oluwaseun Akeju |
| National Institutes of Health | DP1-OD003646 | Emery N Brown |
| National Institutes of Health | TR01-GM104948 | Emery N Brown |
| National Institutes of Health | DP2-OD006454 | Patrick L Purdon |

The funders had no role in study design, data collection and interpretation, or the decision to submit the work for publication.

### Author contributions

OA, Conceived project, Designed experiments, Acquired data, analyzed data, Interpreted data and wrote the manuscript; MLL, Designed experiments, Analyzed data, Interpreted data, Wrote the manuscript; CC, Analyzed data, Interpreted data, Critically revised the manuscript; KJP, Analyzed data, Acquired data, Critically revised the manuscript; RV, JR, GA, SH, KH, Acquired data, Critically revised the manuscript; VCR, DBC, DI-G, Analyzed data, Critically revised the manuscript; JMH, VN, Interpreted data, Critically revised the manuscript; ENB, Conceived project, Designed experiments, Interpreted data, Critically revised the manuscript; PLP, Interpreted data, Wrote the manuscript

### Ethics

Human subjects: The Human Research Committee and the Radioactive Drug Research Committee at the Massachusetts General Hospital approved the study protocol. After an initial email/phone screen, potential study subjects were invited to participate in a screening visit. At the screening visit, informed consent including the consent to publish was requested after the nature and possible consequences of the study was explained. All subjects provided informed consent and were American Society of Anesthesiology Physical Status I with Mallampati Class I airway anatomy.

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
