## [Decision Letter]

Thank you for sending your work entitled “Thalamic Functional Connectivity to the Default Mode Network is a Fundamental Neural Correlate of Consciousness” for consideration at *eLife*. Your article has been favorably evaluated by Eve Marder (Senior editor), Jody Culham (Reviewing editor), and 3 reviewers.

The Reviewing editor and the reviewers discussed their comments before we reached this decision, and the Reviewing editor has assembled the following comments to help you prepare a revised submission.

All three external reviewers made positive comments about the manuscript, noting it “presents an interesting and novel approach to the problem of defining the neural bases of anesthesia-induced unconsciousness”, provide results that are “highly compelling” and “significant”, and overall is a paper that is “quite interesting.”

However, each reviewer brought up points that must be addressed in a revision before the manuscript can be considered suitable for publication in *eLife*. The following points must be addressed in a revision, which should also take into account the secondary comments (taken directly from the reviewers).

1) Although one reviewer acknowledged that the definition of “loss of consciousness” is a matter of semantics, the authors should rewrite the manuscript to focus on what they actually observed – unresponsiveness – and not perseverate on loss of consciousness.

It is possible that the disconnection of the cortico-thalamic-cerebellar cohesion which the authors claim to be the dexmedetomidine-induced perturbation that produces unresponsiveness may be linked to loss of motor output rather than the conscious experience (Tononi et al., 2004). Without studying the volunteers' subjective experience (after the period of sedation) they would be hard-pressed to describe what was studied as loss of consciousness. [60] showed that almost 3 of 4 of their subjects with dexmedetomidine-induced unresponsiveness reported a subjective experience. Absent documentation of presence/absence of subjective experience and therefore presence/absence of consciousness, the possibility remains that most of the subjects retained consciousness and hence cortico–cortico connectivity. A further doubt whether the subjects had lost consciousness is their description of the similarity of the dexmedetomidine-induced behavioral state to EEG-defined stage 2 NREM sleep; this sleep state is not uniformly associated with a lack of subjective experience and therefore unconsciousness.

2) The Introduction should make the main research question and goals clearer. For example, are they using anesthesia as a model to explain pathological loss of consciousness (e.g. disorders of consciousness) or are they interested in the effects of anesthesia in the brain's resting dynamics per se?

3) The revision should discuss/reconcile the decrease in rCBF in DMN, FPN and thalamus with the return of consciousness in the context of the restoration of the functional connectivity in the same areas. One reviewer suggests “perhaps there is some evidence that the alpha2 Dex specific agonism may influence blood vessel responses that could account for both the preservation of the functional connectivity at the cortical level during general anesthesia and failure to normalize?”

4) The spatial resolution of the techniques should be fully considered, particularly in the context of the ability to detect effects in thalamic subnuclei.

5) The authors may wish to take better advantage of the fact that they collected fMRI/PET data simultaneously. All comparisons between both modalities are made on a descriptive basis (or map overlap). A direct comparison of the changes in activation/connectivity/metabolic maps over time would add stronger support to their conclusions.

Reviewer #1:

The authors' reference to the need of combined EEG and fMRI in the Introduction (end of second paragraph) seems out of the scope of their article.

The Methods section describes only 1 imaging visit but Figure 2 describes 2. Please clarify.

Also the number of subjects included in is not clear. Were there 10 subjects for the PET data and 17 or 10 for the fMRI data? Please clarify. Why didn't the authors collect data from all subjects?

How was the onset of the 'unconscious' period determined? In the Methods section the authors define 'sustained eye closure' as a criterion but in the Results section they define an 'EEG response similar to nREM 2 sleep' instead.

The authors may want to consider including a reference to the article below, as it addresses similar research questions:

Altered temporal variance and neural synchronization of spontaneous brain activity in anesthesia.

Huang Z, Wang Z, Zhang J, Dai R, Wu J, Li Y, Liang W, Mao Y, Yang Z, Holland G, Zhang J, Northoff G.

Hum Brain Mapp. 2014 May 28. doi: 10.1002/hbm.22556.

Reviewer #2:

Discussion of the role of noradrenergic projections from locus ceruleus to central thalamus could be amplified and the specific innervation of the central lateral nucleus neurons by noradrenergic afferents noted (Vogt, **B**.A. et al., 2008. Norepinephrinergic afferent and cytology of the macaque monkey midline, mediodorsal, and intralaminar thalamic nuclei. Brain Struct. Funct. 212: 465-479.) This specificity relates to the findings of the relationship of posterior DMN structures and loss of consciousness identified in the manuscript (see point 3).

The observation that thalamic seed is specifically functionally disconnected from posterior DMN regions (posterior cingulate, precuneus and inferior parietal cortex) is relevant to the specific role of the central lateral nucleus components of the intralaminar nuclei which send broad projections across these cortical regions (Buckwalter et al. J. Comp Neurol. 2008). Recent FDG-PET studies in severely brain-injured subject demonstrate a graded relationship between central thalamus and the precuneus (Fridman et al. PNAS 2014) and provide a nice correlation of these findings which could be compared and discussed.

The correlation of the correlation of the Dex-induced unconscious state with the specific preservation of cortico-cortical functional connectivity suggests an interesting model for minimally conscious state. As the authors have noted this literature in the manuscript it might be worth a brief elaboration of the possible implication of these findings for similar states arising after structural brain injuries.

Reviewer #3:

Introduction:

The authors should quote original articles about the mechanism of action of dexmedetomidine, including the neural loci of action rather than their own review (10).

Methods:

The authors mention that principled constraints were placed on the connectivity analysis. These have not been referenced or defined elsewhere. It would be important to clarify.

Results:

Does the [78] method have the spatial resolution to be able to detect activity in thalamic subnuclei? If not did you consider using a voxel-level correction (so that the statistical inference is about the peak voxel, and is localised accurately) or cluster level correction with a more stringent voxel level threshold (so the activations are more focal and better localised, albeit still imperfectly)?

Discussion:

The authors have compared their findings using dexmedetomidine with other author's studies on propofol. Is this comparison justified given the author's claim to have used a novel method to identify the changes in connectivity, an analysis that was not used by the other authors? Is it possible that the Akeju et al.'s methods are not as sensitive as other authors work?

There appears to be tautology associated with the authors contention that slow wave sleep is a “deeper level of unconsciousness” than that provided by dexmedetomidine. This therefore assumes that loss of consciousness (unconsciousness) is *not* binary. Therefore I am confused by their speculation.

The authors do not discuss the finding of reduced CBF on recovery of responsiveness: this is one of the most interesting and perplexing findings that deserves some discussion.

Figure 1 ignores the effect of DEX at postsynaptic alpha2 receptors, which appear to be important for the decrease in anesthetic requirements that is produced by dexmedetomidine (Segal et al., 1988).

---

## [Author Response]

*1) Although one reviewer acknowledged that the definition of “loss of consciousness” is a matter of semantics, the authors should rewrite the manuscript to focus on what they actually observed – unresponsiveness – and not perseverate on loss of consciousness*.

*It is possible that the disconnection of the cortico-thalamic-cerebellar cohesion which the authors claim to be the dexmedetomidine-induced perturbation that produces unresponsiveness may be linked to loss of motor output rather than the conscious experience (Tononi et al., 2004). Without studying the volunteers' subjective experience (after the period of sedation) they would be hard-pressed to describe what was studied as loss of consciousness.*
[60]
*showed that almost 3 of 4 of their subjects with dexmedetomidine-induced unresponsiveness reported a subjective experience. Absent documentation of presence/absence of subjective experience and therefore presence/absence of consciousness, the possibility remains that most of the subjects retained consciousness and hence cortico–cortico connectivity. A further doubt whether the subjects had lost consciousness is their description of the similarity of the dexmedetomidine-induced behavioral state to EEG-defined stage 2 NREM sleep; this sleep state is not uniformly associated with a lack of subjective experience and therefore unconsciousness*.

We thank the reviewers for these comments. We have taken care to rewrite the manuscript without alluding to the notion that we studied consciousness. We agree that the definition of anesthesia-induced loss of consciousness remains a matter of debate in the literature. We have changed the title of this manuscript to reflect that we studied dexmedetomidine-induced unconsciousness:

“Disruption of Thalamic Functional Connectivity is a Neural Correlate of Dexmedetomidine-Induced Unconsciousness”

In our Introduction, we now also explicitly orient the reader as follows:

“We note that precise definitions of the words “consciousness” and “unconsciousness” remain ambiguous. However, loss of voluntary response in a task is an effective and clinically relevant definition of unconsciousness.”

While we agree that alterations in cortico-thalamic-cerebellar networks may in principle reflect loss of motor output, we would like to point out that we did not see evidence of alterations in brain regions associated with motor function. The cerebellum has been traditionally thought to solely contribute to planning and execution of movement, however recent research now demonstrates that the cerebellum is also involved in higher-order neural processing and cognition (Buckner, Neuron 2013 October 30;80(3):807-15). Indeed, the cerebellar regions displaying alterations in connectivity during dexmedetomidine-induced anesthesia in our study did not correspond to the cerebellar representation of the sensorimotor network. Rather they correspond to regions previously described as the cerebellar components of Default Mode Network and the Fronto-parietal networks (Buckner et al., J Neurophysiology 2011 November;106(5):2322-45), which are involved in cognitive processes (including self-referential cognition and attention processes) as opposed to motor processes. At our institution, dexmedetomidine is preferentially used for craniotomies involving cortical motor mapping (masses abutting the motor cortex) due to its favorable “arousability to consciousness” profile. We have not found that dexmedetomidine impairs motor output/motor-evoked potentials (Garavaglia et al., J Neurosurg Anesthesiol. 2014 July;26(3):226-33; Moore et al., Anesth Analg. 2006 May;102(5):1556-8).

The debate on the presence of subjective experiences during anesthesia has been fueled by studies of anesthesia-induced subjective experiences elicited from questionnaires typically administered during the post-operative or post-anesthesia period. This method of investigation does not precisely inform us on the timing of the subjective experience (anesthesia induction vs surgical anesthesia level maintenance vs anesthesia emergence). Nor does it inform us about the timing of that subjective experience relative to the subject’s brain state, which can fluctuate substantially even at a “steady state” of anesthetic administration, and certainly during induction and emergence. In particular, these studies were not performed with an a priori or post hoc analysis of the neurophysiological state(s) elicited by the general anesthetic agent. Rather, the Bispectral Index was typically used as a crude measure of the anesthetized brain. Research and clinical experience from our group and others indicate that the Bispectral Index output varies substantially from clinical assessments of unconsciousness. This is because the output from this device attempts to simplify the complex nature of the electroencephalogram to a single number between 1 and 100. To better inform the readers and to foster more principled approaches to defining and studying unconsciousness/lack of responsiveness in neuroscience, in the revised version of this manuscript, we now describe a few nuances pertaining to anesthesia- and sleep-induced unconsciousness in the context of the subjective experience(s) that have been elicited and positioned to argue against unconsciousness in these states (section titled “Subjective awareness during sleep and anesthesia”).

Therefore, we feel it is appropriate to refer to the state of altered arousal described in this manuscript as dexmedetomdine-induced unconsciousness.

*2) The Introduction should make the main research question and goals clearer. For example, are they using anesthesia as a model to explain pathological loss of consciousness (e.g. disorders of consciousness) or are they interested in the effects of anesthesia in the brain's resting dynamics* per se*?*

We agree with this concern, and we have rewritten our Introduction to more accurately reflect that we are studying anesthesia-induced unconsciousness:

“Therefore, the aim of this study was to use a novel integrated positron emission tomography and magnetic resonance imaging (PET/MR) approach characterize brain resting-state network activity and metabolism during dexmedetomidine-induced unconsciousness.”

3) The revision should discuss/reconcile the decrease in rCBF in DMN, FPN and thalamus with the return of consciousness in the context of the restoration of the functional connectivity in the same areas. One reviewer suggests “perhaps there is some evidence that the alpha2 Dex specific agonism may influence blood vessel responses that could account for both the preservation of the functional connectivity at the cortical level during general anesthesia and failure to normalize?”

We agree that this is a point that should be discussed. We have now added more detailed conjecture for the sustained decrease in rCBF (section titled “Reduced cerebral blood flow during recovery of consciousness”).

*4) The spatial resolution of the techniques should be fully considered, particularly in the context of the ability to detect effects in thalamic subnuclei*.

We agree with the reviewer that the spatial resolution of our techniques should be placed in the appropriate context when reporting results with the thalamic nuclei. We have included the following statement in the Methods section (Decreased thalamo-cortical and cortico-cerebellar functional connectivity during dexmedetomidine-induced unconsciousness):

“However, our ability to precisely resolve the thalamic nuclei implicated in loss of thalamo-cortical functional connectivity is limited because our methods are not sensitive to small focal differences.”

*5) The authors may wish to take better advantage of the fact that they collected fMRI/PET data simultaneously. All comparisons between both modalities are made on a descriptive basis (or map overlap). A direct comparison of the changes in activation/connectivity/metabolic maps over time would add stronger support to their conclusions*.

We agree with the reviewers that a direct comparison of activation, connectivity and metabolic maps over time would be interesting. However, this is not possible given the technical limitations of these methods. While H2O-PET has been used to characterize group-level dynamics and functional connectivity, the FDG-PET we used in our paper estimates glucose consumption over a time-scale of minutes. The technology lacks the temporal resolution to capture physiological alterations associated with faster brain dynamics such as dexmedetomidine-induced brain changes in glucose utilization. Therefore, changes in glucose utilization induced by experimental paradigms such as ours are typically inferred by comparing two separate scans (baseline vs active study scan). While ASL and BOLD fMRI have a faster temporal resolution at the level of several seconds, the SNR of these methods requires pooling of information across many minutes of imaging (in essence, averaging). Therefore, contemporary analyses of fMRI data are almost always limited to static analyses (brain maps) obtained over a several-minute period. Thus, unfortunately, it is not possible to study detailed time-varying activity with any of these imaging modalities.

Reviewer #1:

*The authors' reference to the need of combined EEG and fMRI in the Introduction (end of second paragraph) seems out of the scope of their article*.

We agree and have removed this reference in our revision.

*The Methods section describes only 1 imaging visit but*
Figure 2
*describes 2. Please clarify.*

We apologize for the confusion. We have explicitly added this detail to the Data analysis portion of the Methods section.

“PET data collected from 10 subjects (two visits each) and stored in list mode format were binned into sinograms.”

Also, the caption for Figure 2 has also been amended;

*“*Schematic of fMRI obtained concurrently in combined PET/MR visits (n = 10) and MR/only visits (n = 7)*”*

Also the number of subjects included in is not clear. Were there 10 subjects for the PET data and 17 or 10 for the fMRI data? Please clarify. Why didn't the authors collect data from all subjects?

We regret this. However, in the Results section (Dexmedetomidine-induced unconsciousness decreased CMRglc in the Default Mode Network (DMN), Fronto Parietal Networks (FPNs), and thalamus) we explicitly state the following:

“We then studied ^18^F-FDG uptake during baseline and dexmedetomidine-induced unconsciousness in these 10 healthy volunteers…”

We have added the following sentence to directly follow this sentence in order to provide more context:

“During both the baseline and dexmedetomidine-induced unconsciousness ^18^F-FDG study visits (two per subject), fMRI data were also recorded.”

We hope that these revisions will clarify the situation and address the reviewer’s concerns.

*Please clarify*. *Why didn't the authors collect data from all subjects?*

FDG-PET studies of propofol and isoflurane induced-unconsciousness, the effect sizes observed were large enough to yield statistically significant effects with only six and five volunteers (Alkire MT. et al., Anesthesiology 1997, 1995; Alkire MT. et al.). Based on discussions with experts in the field, and correspondence with our institutional review board the PET portion of our study was appropriately powered (n = 10, which exceeds that of previous studies without unduly exposing study volunteers to unnecessary radiation risks). In addition, the combined PET-fMRI studies are expensive and very labor intensive. Thus, we felt that a sample size of n = 10 would be more than adequate, exceeding previous similar studies, and would not be overly burdensome in terms of subject risk and cost.

Also, in our Methods section we have added the following to better clarify fMRI data collection:

“fMRI data were collected from 17 subjects. However, BOLD data (awake, unconscious, and recovery) were successfully collected in only 15 subjects. This is because BOLD data was not collected in one subject, and another subject exited the scanner immediately prior to the recovery BOLD scan.”

*How was the onset of the 'unconscious' period determined? In the Methods section the authors define 'sustained eye closure' as a criterion but in the Results section they define an 'EEG response similar to nREM 2 sleep' instead*.

We have amended the text in our Methods section.

**“**Volunteers were instructed to keep their eyes open during the course of the study; loss of consciousness was defined as the onset of sustained eye closure and lack of response to a verbal request to open the eyes.”

We have also amended the text in our Results section.

“In 10 healthy volunteers, we used EEG recordings, and an auditory task to confirm that dexmedetomidine induced a loss of voluntary responsiveness, and exhibited a neurophysiological profile that was similar to non-rapid eye movement (nREM) II sleep (1). We then studied ^18^F-FDG uptake during baseline and dexmedetomidine-induced unconsciousness in these 10 healthy volunteers in a separate experiment where we defined loss of consciousness as the onset of sustained eye closure and lack of response to a verbal request to open the eyes (Figure 2).”

The authors may want to consider including a reference to the article below, as it addresses similar research questions:

*Altered temporal variance and neural synchronization of spontaneous brain activity in anesthesia*.

*Huang Z, Wang Z, Zhang J, Dai R, Wu J, Li Y, Liang W, Mao Y, Yang Z, Holland G, Zhang J, Northoff G*.

*Hum Brain Mapp. 2014 May 28. doi: 10.1002/hbm.22556*.

We thank the reviewer for this suggestion. This paper analyzes patterns of resting-state fMRI activity during propofol and sevoflurane-induced unconsciousness. The authors showed that during anesthesia-induced unconsciousness, functional connectivity and temporal variance within the default-mode network decrease, while temporal variance in lateral cortical networks increase, compared to the conscious state. We don’t know how these variance measures relate to more established measures of functional connectivity, nor how the whole thalamic seed functional connectivity relates to our seed based analysis. This is especially important because non-specific and specific thalamic nulei functional connectivity in both sleep and anesthesia are different (Liu et al., Anesthesiology. 2013 January;118(1):59-69; Picchioni et al., Sleep. 2014 February 1;37(2):387-97). Thus we are hesitant to cite it without a specific understanding of how it relates to our results and previous studies.

Reviewer #2:

*Discussion of the role of noradrenergic projections from locus ceruleus to central thalamus could be amplified and the specific innervation of the central lateral nucleus neurons by noradrenergic afferents noted (Vogt,*
***B****.A. et al. 2008. Norepinephrinergic afferent and cytology of the macaque monkey midline, mediodorsal, and intralaminar thalamic nuclei. Brain Struct. Funct. 212: 465-479.) This specificity relates to the findings of the relationship of posterior DMN structures and loss of consciousness identified in the manuscript (see point 3)*.

*The observation that thalamic seed is specifically functionally disconnected from posterior DMN regions (posterior cingulate, precuneus and inferior parietal cortex) is relevant to the specific role of the central lateral nucleus components of the intralaminar nuclei which send broad projections across these cortical regions (Buckwalter et al. J. Comp Neurol. 2008)*.

We thank the reviewer for this suggestion. We have incorporated these insights into the following revised discussion paragraph (“A putative functional network for recovery from unconsciousness comprising the locus ceruleus, central thalamus, and posterior cingulate cortex”).

*Recent FDG-PET studies in severely brain injured subject demonstrate a graded relationship between central thalamus and the precuneus (Fridman et al. PNAS 2014) and provide a nice correlation of these findings which could be compared and discussed*.

We thank the reviewer for this suggestion. We have modified the text in our Discussion to correlate our findings with the graded relationship between the central thalamus and precuneus that has been recently described in severely brain injured patients (“Reduced cerebral blood flow during recovery of consciousness”).

*The correlation of the correlation of the Dex-induced unconscious state with the specific preservation of cortico-cortical functional connectivity suggests an interesting model for minimally conscious state. As the authors have noted this literature in the manuscript it might be worth a brief elaboration of the possible implication of these findings for similar states arising after structural brain injuries*.

We thank the reviewer for this statement. In this manuscript we have attempted to carefully balance our anesthesia-induced findings with potential implications for DOC, sleep and the practice of anesthesia. However, we have rewritten our Discussion section to more explicitly elaborate on the potential implications of our findings. We sincerely hope that the modified text in our Discussion section would satisfy the reviewers concerns. Specifically in “Different states of unconsciousness may reflect a hierarchy of disruption in functional circuits” section of the discussion, we now explicitly state the following:

**“**In the case of sleep, dexmedetomidine-induced unconsciousness, and some minimally conscious states, maintained cortico-cortical functional connectivity likely allows for the cortex to be “primed,” ready to recover from the altered consciousness states with restoration of thalamic functional connectivity, which could be triggered by ascending midbrain and brainstem inputs (69) or direct therapeutic intervention at the thalamic level (67; 77).**”**

Reviewer #3:

Introduction:

*The authors should quote original articles about the mechanism of action of dexmedetomidine, including the neural loci of action rather than their own review (*[10]*)*.

We regret this. We address this concern in this revision.

Methods:

*The authors mention that principled constraints were placed on the connectivity analysis. These have not been referenced or defined elsewhere. It would be important to clarify*.

By principled constraints, we meant that instead of multiple seed based analysis the FDG-PET and ASL fMRI results helped constrain our analysis to the DMN and FPN. We have rewritten the statement to read as follows:

“Third, we use the CMR_glc_ results to inform the brain functional connectivity analysis derived from the Blood Oxygen Level Dependent (BOLD) fMRI signals.”

Results:

*Does the*
[78]
*method have the spatial resolution to be able to detect activity in thalamic subnuclei? If not did you consider using a voxel-level correction (so that the statistical inference is about the peak voxel, and is localised accurately) or cluster level correction with a more stringent voxel level threshold (so the activations are more focal and better localised, albeit still imperfectly)?*

We agree with the reviewer that other methods for statistical thresholding may be more sensitive to focal, rather than distributed effects. However, we feel that our choice of using cluster correction at the standard cluster-forming threshold of z = 2.3 are appropriate for two reasons:

First, and perhaps most importantly, the factor limiting the resolution of our data is the spatial resolution of the scanner itself. It is commonly accepted that an object needs to be 2-3 times the full width at half maximum (FWHM) of the scanner in order to be properly resolved, with limited partial volume effects. The FWHM of the PET camera used in our study -the Siemens Biograph mMR- is 4.3 mm measured at 1 cm from the center of the field of view (Delso et al., J Nucl Med 2011). Therefore, it follows that only objects with a diameter larger than >1 cm would be fully resolved. Since most of the thalamic nuclei have a size that lies below this threshold (Mai, Paxinos and Voss, Atlas of the Human Brain, 2008, Elsevier), we don't feel that we would be able to successfully distinguish different nuclei without a significant amount of partial voluming.

In addition, we expected very widespread changes in our PET and fMRI signals due to the large effect size of our pharmacological-induced unconsciousness experimental model. As a result, we believe that the adoption of our thresholding methods of choice, which is maximally sensitive to widespread signals is appropriate.

Below we present our data from Figure 1 using more stringent cluster forming thresholds and voxel level threshold. Notably, the peak of the voxel-wise correction localized the peaks identified using cluster forming thresholds ( z = 2.3 or 3.1 or 3.3 ).Author response image 1.

Discussion:

The authors have compared their findings using dexmedetomidine with other author's studies on propofol. Is this comparison justified given the author's claim to have used a novel method to identify the changes in connectivity, an analysis that was not used by the other authors? Is it possible that the Akeju et al.'s methods are not as sensitive as other authors work?

We thank the reviewer for this comment. Even though we used a novel PET/MR method to acquire functional and metabolism data in subjects during the same experimental session, the method we used to compute functional connectivity analyses within specific networks (i.e., ‘dual regression’) has been widely used in the literature, and was found to be very sensitive and to have moderate-to-high test–retest reliability. Moreover, the scanning parameters we used generated a slightly higher spatial resolution than most studies (voxel size = 2.3 × 2.3 × 3.8 mm). For these reasons, we feel that the comparison with other connectivity studies in the literature is justified and is not confounded by lower sensitivity in our method.

*There appears to be tautology associated with the authors contention that slow wave sleep is a “deeper level of unconsciousness” than that provided by dexmedetomidine. This therefore assumes that loss of consciousness (unconsciousness) is* not *binary. Therefore I am confused by their speculation*.

We have rewritten this sentence to read as follows:

“However, during the more profound sleep state of nREM III, there is a…”

*The authors do not discuss the finding of reduced CBF on recovery of responsiveness: this is one of the most interesting and perplexing findings that deserves some discussion*.

We thank the reviewer for this comment. We address this concern in the revised version of this manuscript (response to comment 3 of the combined reviews above).

Figure 1
*ignores the effect of DEX at postsynaptic alpha2 receptors, which appear to be important for the decrease in anesthetic requirements that is produced by dexmedetomidine (Segal et al 1988)*.

We regret this omission. In the revised caption, we acknowledge the effect of dexmedetomine at post-synaptic alpha-2-receptors***.***